# Flow Models for Unbounded and Geometry-Aware Distributional Reinforcement Learning

## Abstract

We introduce a new architecture for Distributional Reinforcement Learning (DistRL) that models return distributions using normalizing flows. This approach enables flexible, unbounded support for return distributions, in contrast to categorical approaches like C51 that rely on fixed or bounded representations. It also offers richer modeling capacity to capture multi-modality, skewness, and tail behavior than quantile based approaches. Our method is significantly more parameter-efficient than categorical approaches. Standard metrics used to train existing models like KL divergence or Wasserstein distance either are scale insensitive or have biased sample gradients, especially when return supports do not overlap. To address this, we propose a novel surrogate for the Cramér distance, that is geometry-aware and computable directly from the return distribution's PDF, avoiding the costly CDF computation. We test our model on the ATARI-5 sub-benchmark and show that our approach outperforms PDF based models while remaining competitive with quantile based methods.

## 1 Introduction

Traditional reinforcement learning (RL) algorithms aim to estimate the expected return from a given state or state-action pair (Sutton & Barto, 2018). However, this expectation provides only a partial view of the underlying return distribution, omitting critical information about uncertainty, risk, and variability. Distributional reinforcement learning (DistRL) addresses this limitation by modelling the full return distribution, providing a richer and more informative signal for decision-making.

A key challenge in DistRL lies in how to effectively represent and optimize over return distributions. Early approaches such as Categorical DQN (C51) (Bellemare et al., 2017a) approximate the return distribution with a fixed discrete support. However, this formulation imposes a bounded return range, which significantly limits its applicability to problems with unbounded or highly variable returns—common in robotic control tasks involving long-horizon planning (Eysenbach et al., 2019). Environments with open-ended reward accumulation like Ant or Half-Cheetah in MuJoCo (Todorov et al., 2012) can also benefit from an unbounded return support. Similarly, domains like finance (Charpentier et al., 2020), multi-task learning (Yu et al., 2020), transfer learning (Zhao et al., 2020), or lifelong learning (Khetarpal et al., 2022) require algorithms capable of modelling unbounded or skewed return distributions.

Bellemare et al. (2017a) showed that the Distributional Bellman operator is a contraction over a maximal form of the Wasserstein distance and not for the KL divergence. However, they had to train C51[1] using the KL divergence as the Wasserstein distance has biased sample gradients. Along with a theoretical issue, using the KL divergence is also a major drawback in the unbounded setting, where predicted and target distributions may have negligible overlap, resulting in poor gradient signals.

Subsequent quantile-based methods like QR-DQN (Dabney et al., 2017), IQN (Dabney et al., 2018), FQF (Yang et al., 2019) represent the return distribution via learned quantiles and train using quantile regression losses, effectively approximating Wasserstein distances on sample transitions without bias. While these

---

[1]Rainbow (Hessel et al., 2017) builds on C51 with additional RL improvements but inherits its same categorical limitations

methods removed the need for fixed support, their expressiveness is constrained by their fixed quantile structure, limiting their ability to capture skewed or multimodal distributions effectively. Additionally, Quantile regression methods use the Huber loss which makes distributional estimation guarantees vanish. Indeed, the issue lies in the nonlinearity of the quantile Huber loss and the lack of a known contraction property in the distributional Bellman operator when using quantile approximations with this loss. In Rowland et al. (2024) the authors proved that for Quantile Temporal Difference methods (QTD), the iterates of the QTD algorithm converge almost surely to the set of fixed points of a projected distributional Bellman operator (depending on quantile interpolation), under standard stochastic approximation assumptions. However, while this convergence result is strong and addresses prior theoretical gaps, it is specific to tabular QTD (i.e., no function approximation or deep networks) and shows convergence to a fixed point of a projected operator, not necessarily to the true return distribution. Therefore it does not apply directly to QR-DQN and IQN, which sample from a continuous distribution of quantiles and use function approximation.

Jullien et al. (2024) observed that the estimated distributions rapidly collapse to their mean and proposed to mitigate that issue by learning the return distributions expectiles along with their quantiles. Another drawback of quantile based methods is quantile crossing, although mitigated in (Zhou et al., 2020), but can still occur and violate the monotonicity of the cumulative distribution function.

We propose a distributional RL method (NFDRL) that models return distributions as continuous, unbounded densities using normalizing flows (NF)(Papamakarios et al., 2021). Unlike fixed-bin (C51) or fixed-quantile (QR-DQN) approaches, our model adapts its support and resolution based on data, offering high expressivity with fewer parameters[2]. We optimize using the Cramér distance, which better aligns distributions than KL or Wasserstein metrics. Since our model outputs a PDF (not a CDF), we introduce a surrogate loss based on the PDF alone and prove it retains key theoretical properties.

Our method combines: (1) native support for continuous, unbounded returns; (2) adaptive density allocation; (3) strong parameter efficiency; (4) a principled, interpretable loss function.

We first test our approach on toy MDP setups to show its expressiveness. Then we test our model on the ATARI-5 (Aitchison et al., 2022) sub-benchmark and show that our approach outperforms PDF based models while remaining competitive with quantile based methods.

## 2 Background

We consider the standard RL setting, where the interaction between an agent and an environment is modelled as a Markov Decision Process (MDP) $\mathcal{M} = (\mathcal{X}, \mathcal{A}, R, P, \gamma)$ (Sutton & Barto, 2018). Here, $\mathcal{X}$ and $\mathcal{A}$ denote the state and action spaces, $R : \mathcal{X} \times \mathcal{A} \to \mathbb{R}$ is the reward function, $P(\cdot|x, a)$ is the transition kernel, and $\gamma \in (0, 1)$ is a discount factor. A policy $\pi(\cdot|x)$ maps a state $x \in \mathcal{X}$ to a distribution over actions.

Under a fixed policy $\pi$, the return $G^\pi(x)$ is defined as the random variable representing the discounted sum of rewards collected along a trajectory starting from state $x$:

$$G^\pi(x) := \sum_{t=0}^\infty \gamma^t R(x_t, a_t), \quad x_0 = x, a_t \sim \pi(\cdot|x_t), x_{t+1} \sim P(\cdot|x_t, a_t). \tag{1}$$

The goal is to estimate the expected return i.e. the value function $V$ or the action-value function $Q$:

$$V^\pi(x) := \mathbb{E}[G^\pi(x)] = \mathbb{E}\left[\sum_{t=0}^\infty \gamma^t R(x_t, a_t) \Big| x_0 = x\right].$$

$$Q^\pi(x, a) := \mathbb{E}[G^\pi(x, a)] = \mathbb{E}\left[\sum_{t=0}^\infty \gamma^t R(x_t, a_t), \Big|, x_0 = x, a_0 = a\right].$$

---

[2]Our method supports discrete actions; continuous-action settings like D4PG are out of scope here.

These quantities satisfy the classical Bellman equation:

$$Q^\pi(x, a) = \mathbb{E}[R(x, a)] + \gamma \mathbb{E}_{X' \sim P(\cdot|x,a), A' \sim \pi(\cdot|X')}[Q^\pi(X', A')]. \tag{2}$$

**Distributional Reinforcement Learning**

Rather than approximating only the expectation of returns, DistRL aims to model the entire distribution of $G^\pi(x)$, capturing richer information such as variance, multimodality, or higher moments. This perspective has been shown to improve both learning dynamics and empirical performance (Bellemare et al., 2017a).

We adopt the formulation proposed by Bellemare et al. (2023), which distinguishes between the random return $G^\pi(x)$ and its law $\eta^\pi(x)$, defined as:

$$\eta^\pi(x)(S) := \mathbb{P}(G^\pi(x) \in S), \quad \forall S \subseteq \mathbb{R}. \tag{3}$$

To express the Bellman recursion at the level of distributions, they introduce the concept of a pushforward function. Given a function $f : \mathbb{R} \to \mathbb{R}$ and a probability distribution $\eta$, intuitively we have, if $Z \sim \eta$, then $f(Z) \sim f_\# \eta$.

Let the bootstrap function defined as $b_{r,\gamma} : \mathbb{R} \to \mathbb{R}$ by $b_{r,\gamma}(z) := r + \gamma z$, for fixed $r \in \mathbb{R}$ and $\gamma \in (0, 1)$. Then the distributional Bellman equation over distributions can be written as:

$$\eta^\pi(x) = \mathbb{E}_{a \sim \pi(\cdot|x), X' \sim P(\cdot|x,a)}[(b_{R(x,a),\gamma})_\# \eta^\pi(X')]. \tag{4}$$

This defines the distributional Bellman operator $\mathcal{T}^\pi : \mathcal{P}(\mathbb{R})^{\mathcal{X}} \to \mathcal{P}(\mathbb{R})^{\mathcal{X}}$, where:

$$(T^\pi \eta)(x) := \mathbb{E}_{a \sim \pi(\cdot|x), X' \sim P(\cdot|x,a)}[(b_{R(x,a),\gamma})_\# \eta(X')]. \tag{5}$$

This approach provides a compact, principled notation for distributional TD methods we use.

**Normalizing Flows**

NF are a class of generative models that transform a simple base distribution into a more complex one via a sequence of smooth, invertible mappings (Papamakarios et al., 2021). Starting from a base sample $z_0 \sim p(z_0)$, a flow applies a sequence of transformations $f_k \circ \cdots \circ f_1$ to obtain $z_K$, whose density is computed using the change of variables formula:

$$\log p_\theta(z_K) = \log p(z_0) - \sum_{k=1}^{K} \log \left| \det \left( \frac{\partial f_k}{\partial z_{k-1}} \right) \right|. \tag{6}$$

This allows flows to model flexible densities while retaining exact likelihoods and differentiability.

## 3 Normalizing Flows for Distributional RL

We present our NF-based model for return distribution estimation. While normalizing flows can map in either direction—base to return distribution or vice versa—RL constraints guide this choice. We first discuss both options and justify the appropriate direction, then describe the model architecture and how the distributional Bellman operator is implemented using pushforward distributions.

### 3.1 Modeling the Forward Flow: Sampling Returns and Evaluating Densities

Let $\mathcal{U}$ be a base distribution with full support over $\mathbb{R}$ (e.g., a standard Gaussian), and let $z \sim \mathcal{U}$. Let $f$ be a diffeomorphism from $\mathbb{R}$ to itself (i.e., a bijective, differentiable mapping with a differentiable inverse).

The flow $f$ can be interpreted in two equivalent ways:

- either as mapping return outcomes $y \sim \eta^{\pi}(x, a)$ into the base space: $z = f(y)$,
- or as mapping base samples $z \sim \mathcal{U}$ to return outcomes via: $y = f(z)$, where $y \sim \eta^{\pi}(x, a)$.

Mapping from return outcomes to base samples (i.e., modeling the inverse flow) is attractive because it allows direct likelihood evaluation of observed returns, which is beneficial for maximum likelihood training. However, in reinforcement learning, we do not observe true return samples $y \sim \eta^{\pi}(x, a)$ directly. We must construct them through bootstrapping. This makes the inverse mapping impractical in our context. Instead, we opt to model the forward flow, transforming base noise samples $z \sim \mathcal{U}$ into return outcomes $y = f_{\theta}(z)$, which allows us to generate samples from $\eta^{\pi}(x, a)$ and define pushforward distributions suitable for implementing the distributional Bellman operator.

While choosing to represent return samples as $y = f_{\theta}(z)$, with $z \sim \mathcal{U}$, makes sampling straightforward, it is also possible to compute the probability density of a given return value using the change of variable formula. Let $f_{\theta} : \mathbb{R} \rightarrow \mathbb{R}$ be a flow parameterized by $\theta$, and $p_{\mathcal{U}}$ the density of the base distribution $\mathcal{U}$. Then for $y = f_{\theta}(z)$, we have:

$$\log \eta^{\pi}(x, a)(y) = \log \left( p_{\mathcal{U}}(z) \left| \frac{\partial y}{\partial z} \right|^{-1} \right) = \log \left( p_{\mathcal{U}}(z) \left| \frac{\partial f_{\theta}(z)}{\partial z} \right|^{-1} \right) = \log p_{\mathcal{U}}(z) - \log \left| \frac{\partial f_{\theta}(z)}{\partial z} \right| \quad (7)$$

This formulation yields a closed-form expression for the learned density. Computing the log-density of sampled returns requires only the base log-density and the derivative of the flow function. Therefore, choosing a flow function with easily computable derivatives is crucial.

### 3.2 A CDF-Based Flow Architecture for Conditional Return Modeling

We propose an architecture in which a conditional flow function, constructed from a CDF, maps base samples to return values. Since return distributions $\eta^{\pi}(x, a)$ are one-dimensional, we design a 1D flow $F_{\theta}(x, a) : \mathcal{Z} \rightarrow \mathbb{R}$, where $z$ is drawn from a fixed base distribution, and $y = F_{\theta(x,a)}(z)$ is a return sample conditioned on the state-action pair $(x, a)$.

To model this conditional transformation, we use a neural network $h_{\theta}$ that takes as input the state $x$ and outputs, for each discrete action $a_j \in \mathcal{A}$, the parameters of a Gaussian mixture distribution: $\{(w_i^{(j)}, \mu_i^{(j)}, \sigma_i^{(j)})\}_{i=1}^{n}$. These parameters define a mixture CDF, and with $z \sim \mathcal{U}$, we get:

$$y^{a_j} = F^{(a_j)}(z) = \sum_{i=1}^{n} w_i^{(j)} \Phi \left( \frac{z - \mu_i^{(j)}}{\sigma_i^{(j)}} \right)$$

**Why CDF-based flows?** This design offers several advantages:

1. **Tractable Density Computation**: Given $y = F_{\theta}(z)$ with $z \sim \mathcal{U}$, the log-density of $y$ is computed via equation equation 7. Since $\mathcal{U}$ is simple (e.g., standard Gaussian) and $\partial F_{\theta}(z)/\partial z$ is the PDF of a Gaussian mixture, both terms are closed-form.
2. **Efficient Inversion**: Evaluating $p(y)$ requires $z = F_{\theta}^{-1}(y)$. Although Gaussian mixtures lack analytical quantile functions, $F_{\theta}$ is monotonic and differentiable, allowing fast, stable inversion via binary search in 1D.
3. **High Expressivity**: CDFs allow flexible, nonlinear modeling without strong constraints. In contrast, many flow types face limitations: affine flows are rigid unless deeply stacked (Dinh et al., 2015), squared flows require constrained parameters (Ziegler & Rush, 2019), and splines introduce discretization via binning (Müller et al., 2019).

In contrast, our CDF-based approach naturally supports smooth, continuous outputs and avoids discretization altogether. It is conceptually closest to the method in (Ho et al., 2019), which uses a logistic mixture CDF composed with an inverse sigmoid. Like ours, it supports expressive, invertible transformations without sacrificing differentiability.

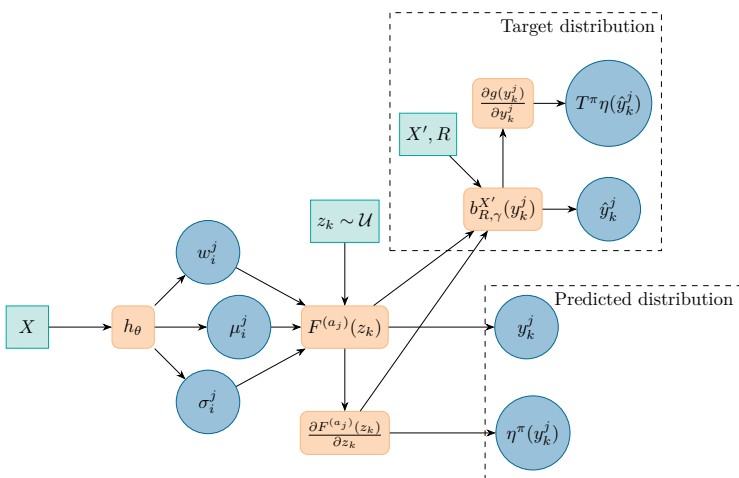

Figure 1: Architecture of the conditional flow model. A neural network $h_\theta$ maps each state $x$ to parameters $\{(w_i^j, \mu_i^j, \sigma_i^j)\}_{i=1}^n$ for each action $a_j \in \mathcal{A}$, defining a mixture CDF $F^{(a_j)}$. This CDF acts as a flow that transforms base noise samples $z_k \sim \mathcal{U}$ into return samples $y_k^j = F^{(a_j)}(z_k)$. The change of variable formula then makes use of the flows derivative to approximate the return distribution $\eta^\pi(y_k^j)$. To estimate the target distribution, the bootsrap function is implemented as a flow and takes the reward and next state as input. It finally outputs target distribution $T^\pi \eta$.

## 3.3 Rescaling the Flow Output: Addressing the Bounded Support of CDFs

Because the chosen flow function is based on a CDF, its output is restricted to $[0, 1]$. However, one of our goals is to allow the predicted return distribution to be continuous and unbounded, reflecting the nature of real-world return values in RL. To overcome this limitation, we introduce an additional transformation step that rescales the output of the CDF-based flow.

$h_\theta$ is extended to output an additional parameter $G_{(x,a)}^{\max}$, which defines the upper bound of the return range for a given state-action pair. This allows the model to flexibly adapt the support of the predicted distribution without manual specification. In this case, we apply an affine transformation to map the CDF output $y \in [0, 1]$ to the desired return range: $f(y) = 2 \cdot y \cdot G^{max} - G^{max}$. The function $f$ is considered as a flow function that comes after the first CDF flow $F$.

Considering that we are now composing two flows ($F$ and $f$) and as per equation 6, this transformation introduces an additional Jacobian term in the log-likelihood of equation 7:

$$\log \eta^\pi(x, a)(y) = \log p_{\mathcal{U}}(z) - \log \left| \frac{dF_\theta(z)}{dz} \right| - \log |2 \cdot G^{max}|. \tag{8}$$

## 3.4 Constructing the RL Target Distribution

Before proceeding, we clarify terminology: in normalizing flows, the target distribution is the flow's output. In our RL setting, this is the predicted return distribution for a state-action pair. During training, it is compared to a separate RL target distribution, which serves as the learning signal. Unless otherwise noted, target distribution refers to this RL target.

We begin by reviewing how target distributions are commonly constructed in DistRL. We then show why directly applying existing methods to our flow-based model does not yield a valid distribution. Finally, we propose a principled solution by introducing a *target flow* that enables a coherent learning objective within our framework.

As shown in equation 5, the Bellman operator $T^\pi$ applies the bootstrap function $b_{r,\gamma}(y) = r + \gamma y$ to samples $y = F_\theta(z)$. However, this scaling distorts probability mass, breaking the change-of-variables formula. **Our**

**key contribution is to treat** $b_{r,\gamma}$ **as a flow layer**. Being affine and invertible, it integrates naturally into the model, preserving normalization. The full composed flow then yields a valid target distribution under the change-of-variable formula.

Let $F_{x',a'}$ denote the flow that predicts the return distribution for the next state-action pair, and let $\tilde{y} = g(y) = r + \gamma y$ be the bootstrap flow. We now have a composition of 3 flows ($F$, $f$ and $g$) and applying equation 6, The target log-density becomes:

$$\log T^{\pi}\eta(x,a)(\tilde{y}) = \log p_z(z) - \log\left|\frac{\partial F_{x',a'}(z)}{\partial z}\right| - \log|2 \cdot G^{max}| - \log\left|\frac{\partial g(F_{x',a'}(z))}{\partial F_{x',a'}(z)}\right| \tag{9}$$

$$= \log p_z(z) - \log\left|\frac{\partial F_{x',a'}(z)}{\partial z}\right| - \log|2 \cdot G^{max}| - \log(\gamma). \tag{10}$$

This approach guarantees that the target distribution is properly normalized, thanks to the compositionality of flows and the use of the change-of-variable formula. Additionally, the extra terms $\log(\gamma)$ and $\log|2 \cdot G^{max}|$ are constant with respect to $z$ and do not incur computational overhead.

Applying the target flow alone does not yield a complete RL target distribution suitable for training, as the predicted distribution $\eta^{\pi}(x,a)$ and the target distribution $T^{\pi}\eta(x,a)$ are defined over different supports. As in the C51 algorithm, which requires projecting the Bellman update onto a fixed categorical support, we must ensure alignment between the two distributions to enable valid comparisons. Hence, we introduce an alignment procedure based on kernel density estimation (KDE). More specifically, we use a KDE to evaluate $\eta^{\pi}$ on the support of $T^{\pi}\eta$, $\tilde{y}$, this is denoted as $\hat{\eta}^{\pi}$. The same is done for $T^{\pi}\eta$ evaluated on the support of $\eta^{\pi}$ (denoted $\hat{T}^{\pi}\eta$)[3].

This two-way evaluation enables us to approximate a symmetric divergence such as the Jensen-Shannon divergence or the Cramér distance between the two distributions.

$$\mathcal{L}(\eta^{\pi}(x,a), T^{\pi}\eta(x,a)) = \sum_{i=1}^{N} D(\eta^{\pi}(x,a)(y_i)||\hat{T}^{\pi}\eta(x,a)(y_i)) + \sum_{j=1}^{M} D(\hat{\eta}^{\pi}(x,a)(\tilde{y}_j)||T^{\pi}\eta(x,a)(\tilde{y}_j))$$

**Final state Gaussian reward approximation** In standard TD methods, the value target at the final time step is simply the immediate reward $r$. In the distributional RL setting, however, we require a full target distribution. Since $r$ is a scalar, it can be viewed as a degenerate distribution — a Dirac delta centered at $r$. To enable learning with continuous distributions, we approximate this Dirac using a Gaussian $\mathcal{N}(r, \sigma)$.

The alignment process is illustrated in the appendix figure 8 and detailed in appendix algorithm 1.

### 3.5 Using the Cramér Distance as a Loss Function

Unlike C51, which uses KL divergence, our method operates on unbounded supports where KL is ill-suited. KL fails when predicted and target distributions do not overlap—common early in training—and lacks translation sensitivity, making it ineffective for disjoint supports. While the Bellman operator is a contraction under the maximal form of the Wasserstein metric (Bellemare et al., 2017a), this metric is hard to optimize with stochastic gradients. Quantile regression (Dabney et al., 2017) addresses this for the 1-Wasserstein case but requires learning quantile values, incompatible with our full-PDF approach. Instead, we use the Cramér distance, which is translation-sensitive, invariant to mass-preserving transforms while providing unbiased sample gradients—making it ideal for our setting (Bellemare et al., 2017b). Definitions and key properties are recalled in the appendix.

Given two CDFs of denoted as $P(x)$ and $Q(x)$, the Cramér distance is defined as follows:

$$C_p(P, Q) = \left(\int_{-\infty}^{\infty} |P(x) - Q(x)|^p dx\right)^{1/p} \tag{11}$$

---

[3]This an abuse of notation as $\eta^{\pi}$ and $\hat{\eta}^{\pi}$ are the same distribution but evaluated on a different set of points. However as both are obtained using different processes; one is the direct output of the model and the other is obtained using a KDE, we use this notation to mark the difference.

Computing the Cramér distance necessitates the CDFs of the predicted and target distributions which are not available. While using the available PDFs of the predicted and target distributions it is possible to compute the CDFs and compute the exact Cramér distance, the involved sorting operation on returns generates computational overhead and becomes tricky or ill-defined in higher dimensions, because multivariate CDFs are not straightforward, and ordering does not generalize well.

We propose a surrogate for the Cramér distance that can be directly computed using the PDFs of the distributions and can be seen as a kernelized $L^2$ norm on PDF differences:

$$D(\eta^\pi(x,a), \hat{T}^\pi \eta(x,a)) = \frac{1}{N^2} \left( \sum_{i,j} \left( \eta^\pi(x,a)(y_i) - \hat{T}^\pi \eta(x,a)(y_i) \right)^2 \cdot |y_i - y_j|^2 \right)^{1/2} \tag{12}$$

This formulation retains the scale sensitivity of the original Cramér distance and allows for stochastic optimization with unbiased gradients. In Appendix 7.3, we provide detailed derivations and prove that this surrogate acts as a valid distance function and preserves contraction properties under the distributional Bellman operator under its maximal form.

## 4 Results

In this section, we empirically validate the proposed DistRL method that we call NFDRL. We first illustrate its ability to model return distributions with tunable variance, showcasing the flexibility offered by our parameterization and the effect of surrogate optimization. We then investigate more complex settings, demonstrating the ability to learn multi-modal distributions. A particular focus is placed on a simple MDP taken from (Jullien et al., 2024) with a bimodal final state distribution (see Appendix figure 11), where quantile-based methods struggle to learn smooth distributions. In contrast, our method successfully handles this challenge.

Building on these controlled experiments, we evaluate our method on a discrete stochastic environment, FrozenLake, to better visualize the multimodal distributions output by our model. We then scale to larger benchmarks by reporting results on a selection of Atari 2600 games, establishing the practicality of our approach in deep reinforcement learning settings.

Finally, we compare C51 and NFDRL model size and provide an empirical analysis of the sample size from base distribution to ensure effective training.

### 4.1 Modeling Expressiveness using simple MDPS

To evaluate the expressiveness of our method, we design three simple MDPs (see Appendix Figure 10).

**MDP 1** The first MDP consists of three states, each with one action. It demonstrates that the learned return distributions can achieve tunable granularity, controlled by two hyperparameters: (i) the standard deviation of the final state's reward distribution, and (ii) the KDE bandwidth during training. Results (Figure 2) show that adjusting these parameters narrows the distributions around their mean, with our method approximating Dirac distributions when trained with the exact Cramér loss. The surrogate loss leads to slightly wider distributions but still offers similar behavior. We believe this is due to the fact that PDFs only reflect local density at a point, not the cumulative mass up to that point. If the predicted and target distributions are close but slightly shifted, their PDFs may not overlap at all. In this case, the surrogate sees a large difference but gives no direction for aligning them (the gradient vanishes or is uninformative). In contrast, CDFs accumulate mass and will differ everywhere after a mismatch, giving strong, global gradients to correct misalignment.

This flexibility contrasts with existing methods: (1) C51 tends to learn broader distributions when rewards fall between fixed atoms; (2) Quantile-based methods (QR-DQN, IQN) represent distributions as sums of Dirac masses at fixed quantiles, lacking control over local spread.

Our method allows smooth interpolation between sharply peaked and broadly spread distributions, combining discrete and continuous representations. We also show in figure 3 the distributions learnt for $(s_1, a_1)$ in $MDP_1$ and that our target flow accurately learns the bootstrap function.

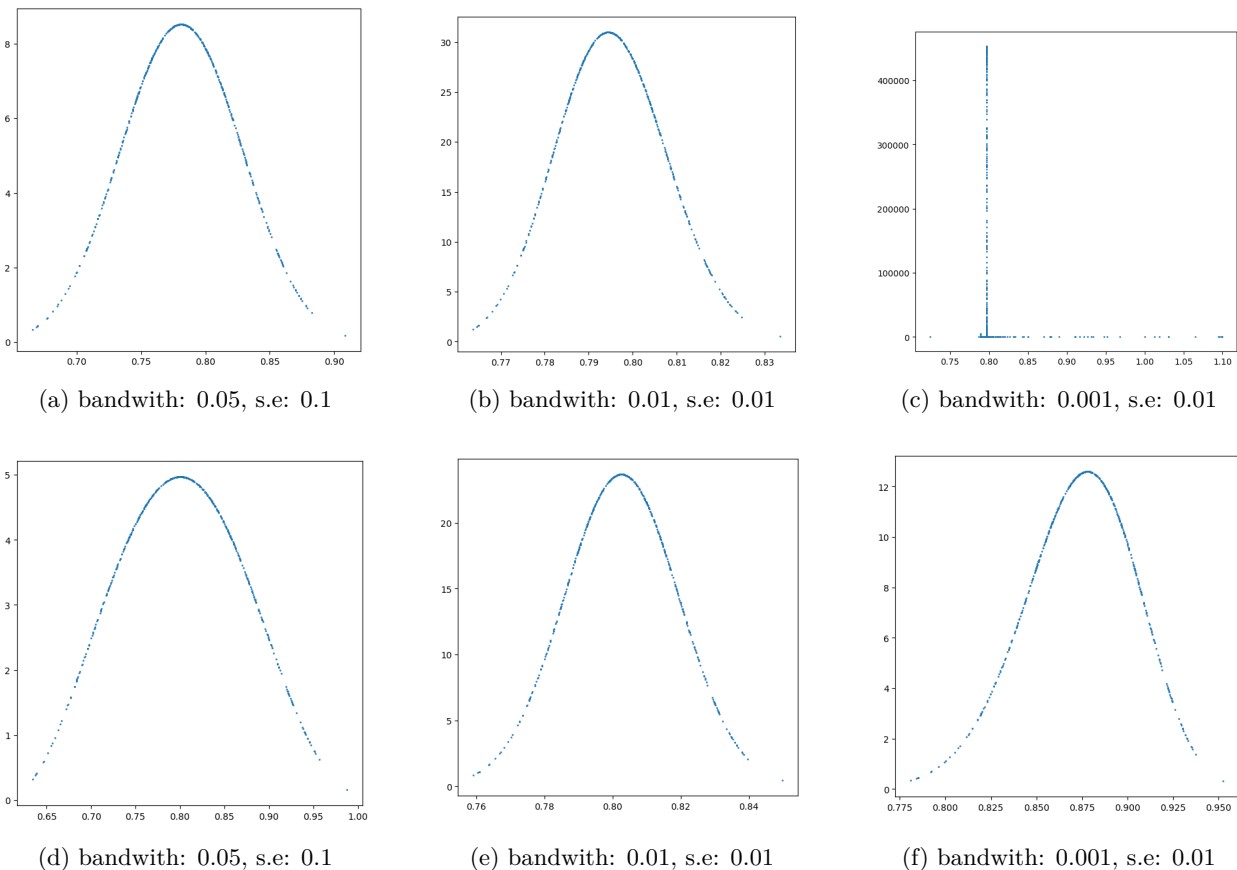

Figure 2: Learnt return distributions for the state-action pair $(s_2, a_1)$ in $MDP_1$, under different values of the KDE bandwidth and final state's reward variance. The x-axis is return values and the y axis corresponds to their corresponding densities. The target reward is 0.8. The top row shows distributions learned using the exact Cramér loss, while the bottom row shows those obtained with our surrogate. Narrower distributions can be achieved with both losses, illustrating the method's flexibility.

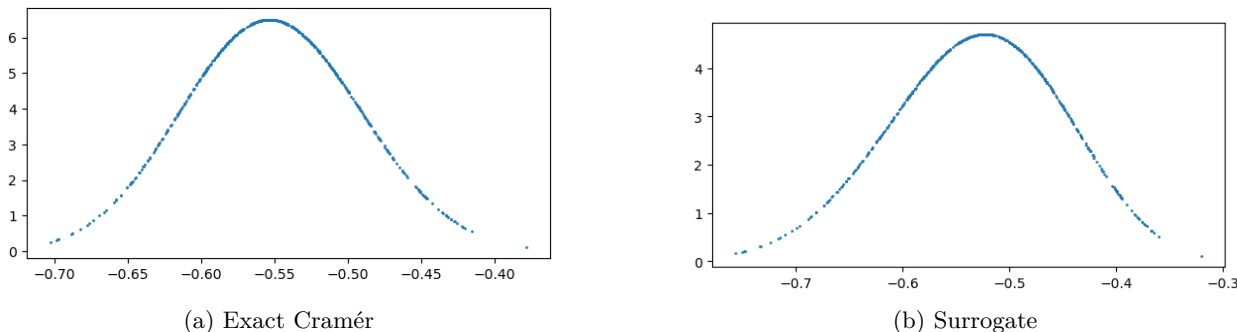

Figure 3: Learnt distributions for $(s_1, a_1)$ in $MDP_1$; $R_1 = -0.8$, $R_2 = 0.3$; $\gamma = 0.9$. The x-axis is return values and the y axis corresponds to their corresponding densities. Here we use a KDE bandwidth of 0.05 and a final reward standard error of 0.01.

**MDP2**   This MDP displays stochastic dynamics with the same action leading to 2 different states with different rewards (Appendix figure 10). Our model can learn multi-modal distributions from samples estimates either using the exact Cramér computation or the surrogate (figure 4).

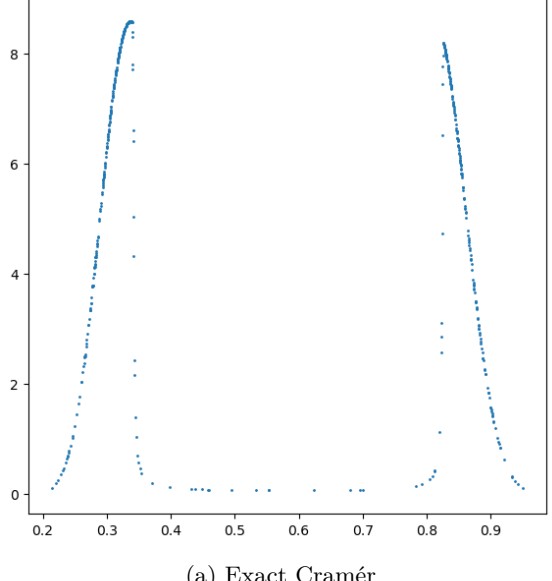 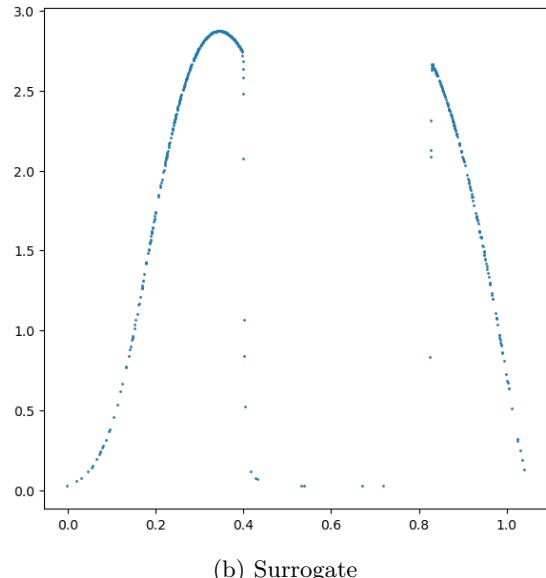

(a) Exact Cramér                                         (b) Surrogate

Figure 4: Learnt distributions for $(s_1, a_1)$ in $MDP_2$; $R_1 = 0.8$, $R_2 = 0.3$. The x-axis is return values and the y axis corresponds to their corresponding densities. We use a KDE bandwidth of 0.05 and a final reward standard error of 0.1

**MDP3**   We reproduce the MDP described in (Jullien et al., 2024) consisting of 4 successive states with only one possible action (Appendix Figure 11). The reward is nil for all states except the last where $R \sim \left(\frac{1}{2}\mathcal{N}(-2, 1) + \frac{1}{2}\mathcal{N}(+2, 1)\right)$. Figure 5 displays the distributions learnt for that final state return using IQN and NFDRL. As noted in (Jullien et al., 2024), quantile regression approximates the inverse CDF with high variance, especially at extremes, leading to noisy and blurred PDF estimates that obscure bimodality. In contrast, our method produces smooth, clearly bi-modal PDFs, albeit with sharper peaks.

**Frozen Lake**   To further assess the expressiveness of our model, we evaluate it on the Frozen Lake environment (Towers et al., 2024), a grid-world domain with inherent randomness. In the Appendix figure 12, we display the learned return distributions for each state-action pair. The environment's stochasticity naturally induces multimodal and skewed return distributions, which our model captures accurately.

## 4.2   Benchmarking

**Environments**   We opted to conduct our experiments with the Atari Learning Environment (ALE) (Machado et al., 2018). In order to accommodate for limited computing resources, we constrained ourselves to the Atari-5 sub-benchmark (Aitchison et al., 2022). As is common with ALE, we also report human-normalized scores.

Implementation details can be found in the appendix (section 7.6).

**Results**   We evaluate our proposed method, NFDRL, in both its exact (NFDRL-E) and surrogate (NFDRL-S) variants across five Atari games, comparing against established baselines including DQN, C51, and IQN. Performance is measured using human-normalized scores (table 1), computed based on the raw game scores of human players and random agents (Appendix table 2). Both NFDRL variants significantly outperform traditional baselines (DQN and C51) on all games, achieving mean scores of 394 (NFDRL-E) and 407

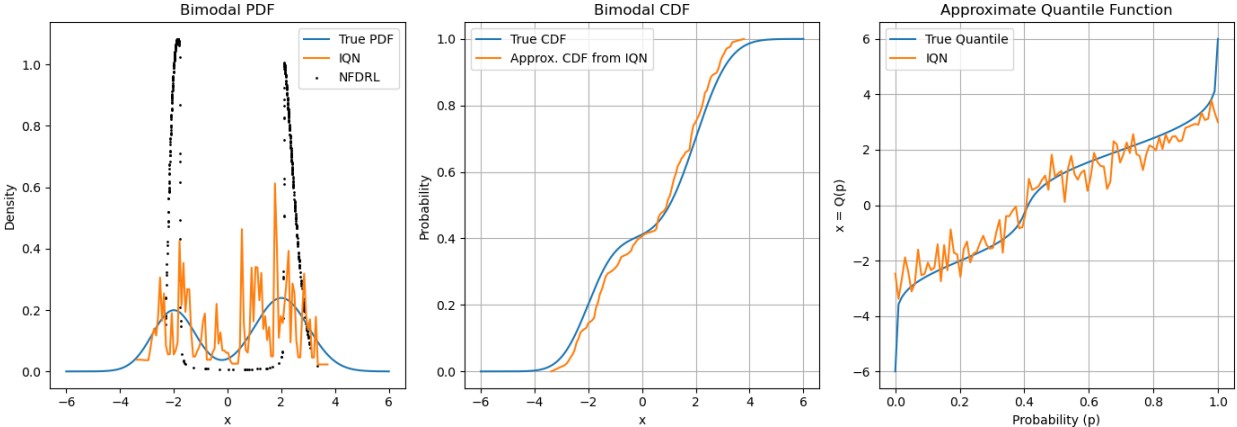

Figure 5: Return distributions learnt for the final state of $MDP_3$ using IQN and NFDRL. **Right**: true quantile function (blue) and quantile function learnt using IQN, reproducing the results of (Jullien et al., 2024). **Middle**: True CDF and CDF approximated for the quantile function obtained from IQN. **Left:** True PDF, and approximations obtained using IQN and NFDRL. While IQN produces a noisy distribution blurring the models, our method outputs a smoother distribution that makes the modes apparent.

(NFDRL-S) versus 189 (DQN) and 320 (C51). Notably, NFDRL-S, which uses a surrogate Cramér loss, slightly outperforms the exact version on average, highlighting its favorable tradeoff between computational efficiency and performance.

Although NFDRL-S uses a surrogate Cramér loss, its slightly better performance compared to the exact version is consistent with trends in deep learning. The exact Cramér loss introduces sampling noise, discretization errors, and sensitivity to distribution tails, which can destabilize training. In contrast, the surrogate loss may better handle these challenges, implicitly reweighting or smoothing the distribution, making it more amenable to optimization. This reflects a broader pattern in deep learning where approximations can lead to better generalization and more stable training, as seen in areas like computer vision with perceptual losses.

While IQN remains the top-performing method overall with a mean score of 525, our method performs competitively, particularly in games such as Double Dunk and QBert*, where NFDRL-S achieves or exceeds IQN performance. In sum, these results validate that the Cramér-based approach is a powerful alternative to quantile-based distributional RL, capable of modeling complex return distributions while maintaining strong empirical performance.

Table 1: Human-normalized performance on ATARI-5. The best performing agent for each game is highlighted in blue. We also compare our approach directly with C51 as they are both approximating the PDF (best one written in bold font) while IQN is quantile based.

| Games | DQN | C51 | IQN | NFDRL-E | NFDRL-S |
|---|---|---|---|---|---|
| Battle Zone | 79 | 76 | **115** | 87 | **95** |
| Double Dunk | 545 | 959 | 1100 | 1200 | **1243** |
| Name this Game | 103 | 178 | **354** | 232 | **259** |
| Phoenix | 119 | 258 | **862** | 280 | **301** |
| Q*Bert | 97 | 178 | 193 | 169 | **193** |
| **Mean** | 189 | 330 | **525** | 394 | **418** |

### 4.3 Parameter efficiency

We assess parameter efficiency by comparing model size and performance to C51. As shown in Figure 6 (left), C51's parameter count grows with the number of atoms, while ours stays constant regardless of the number of components. Our model matches C51's size with just 11 atoms, underscoring its superior efficiency.

We also study the effect of sample count from the base distribution in MDP2. As more samples are used, the Cramér distance to the true distribution steadily drops, with performance stabilizing around 100 samples—offering a good trade-off between accuracy and cost (Figure 6 (right)).

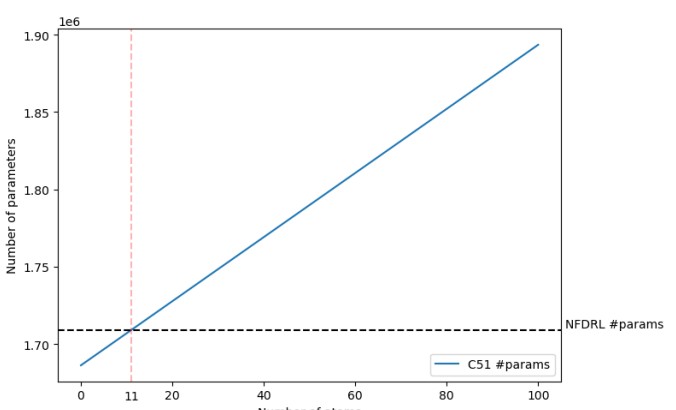

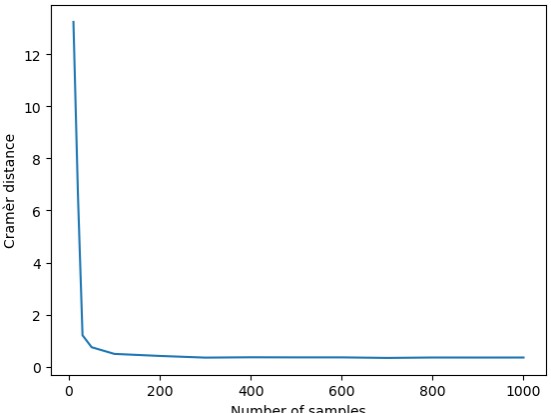

**Parameter count comparison between C51 and NFDRL**. C51's parameter count increases with the number of atoms due to its linear output layer. NFDRL maintains a fixed number of parameters, matching C51 only when it uses 11 atoms, thus demonstrating superior parameter efficiency.

**Impact of the number of base distribution samples on performance**. As the number of samples used for computing the distributional loss increases, NFDRL's loss decreases and performance improves. A plateau is reached around 100 samples, indicating a good trade-off between accuracy and efficiency.

Figure 6: NFDRL Parameter Efficiency

## 5 Conclusion

We introduced a new DistRL method that models return distributions as mixtures of Gaussians, with parameters learned via normalizing flows. By optimizing the Cramér loss—exactly or through a surrogate—we capture richer uncertainty and learn more precise value distributions.

Empirically, our method achieves competitive or better performance on Atari games while being far more parameter-efficient than C51 and more expressive than quantile-based methods.

We focused on flows with CDF-based transformations, but future work could explore alternative architectures, including diffusion-based approaches such as score-based or flow matching models, for added expressiveness and stability. Another promising extension is risk-aware learning using distortion risk measures. This can be achieved by having the model predict both the PDF and CDF of return distributions. Thanks to the invertibility of normalizing flows, the CDF flow can be analytically inverted to obtain the quantile function at negligible cost. This enables the direct use of distortion risk measures facilitating the learning of risk-sensitive policies. Moreover, jointly learning the CDF and PDF may also simplify the computation of the exact Cramér distance, while preserving the advantages of modeling the return distribution's full density.

## 6 Limitations

**Variance**   Our method exhibits high training variance, particularly on games such as Pong. This sensitivity can be partially mitigated by carefully tuning the learning rate schedule, a hyperparameter to which our

model is notably sensitive. Figure 7 illustrates a training run using a fixed learning rate: while the model eventually achieves high performance, its learning curve remains highly unstable. Furthermore, because our loss function relies on sampling from base distributions (e.g., for computing the Cramér distance), it introduces additional stochasticity that contributes to this variance.

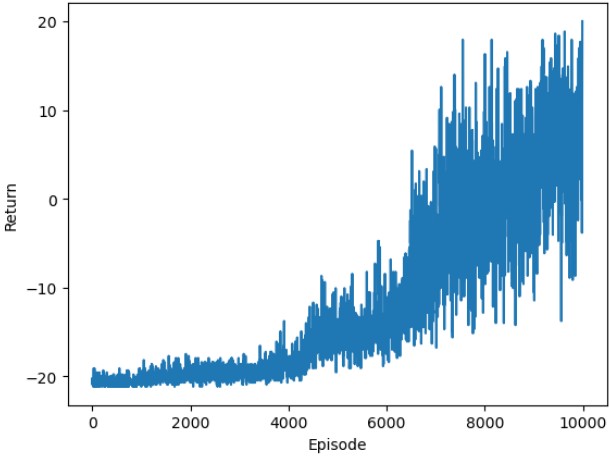

Figure 7: Training curve on PONG without learning rate decay

**Learning efficiency**   As shown in table 1 and figure 7, our model achieves competitive performance but at the cost of a slow training convergence. We believe this can be due to different factors:

1. The training variance might not help the model converge faster

2. Instead of learning directly the density of given values like C51, or specified values, our model learns flow parametrisations that indirectly lead to return distributions. This indirect relationship might hinder the learning performance by making the task more complex for the model.

3. Normalizing Flows are effective for learning exact likelihoods but they are notoriously slow to train, this fact is confirmed by our empirical results.

**CDF Flow**   While using a CDF as a flow transformation offers advantages in modeling monotonic mappings and enabling efficient computation of the Cramér distance (Section 3.2), it also introduces notable limitations. First, the CDF is inherently bounded, making it ill-suited for modeling unbounded return distributions without an additional projection step to extend its support. Second, since the model parameterizes a mixture of Gaussians, the components can become disjoint—e.g., with widely separated means and variances—which may result in poor overlap with the base distribution (typically standard Gaussian). Consequently, this can lead to inefficient coverage of the learned CDF's support, introducing instability and training inconsistencies. Although chaining multiple flow transformations or adopting alternative flow families may alleviate this issue, doing so increases model complexity and may hinder convergence. Addressing this trade-off remains an open direction for future work.

**KDE for Target Distribution**   To enable the computation of the Cramér distance, we construct a target return distribution via KDE, ensuring that it shares the same support as the predicted distribution. However, this introduces a non-trivial computational burden and a non-trainable step in the pipeline, as gradients do not flow through the KDE. We attempted to mitigate this by reusing the same set of samples $z$ (used to parameterize the predicted distribution) to build the KDE target, in the hope of improving alignment and efficiency. Unfortunately, this strategy did not yield significant improvements. Ideally, a more integrated approach would avoid the need for KDE altogether, enabling fully end-to-end training and reducing reliance on handcrafted alignment procedures.

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

# 7 Appendix

## 7.1 Alignment Procedure and Algorithm

As mentionned in section 3.4, our method necessitates a support alignment between the predict and target distributions. Indeed for the same samples $z_i \sim \mathcal{U}$, we have $F_{(x,a)}(z_i) = y_i$ and the target distribution is based on subsequent states and actions, therefore $\tilde{y}_i = F_{(x',a')}(z)$. Said otherwise, as the predicted and target distributions are based using different state-action pairs, the output return values are different are also different, without even taking the bootstrap function into account. Therefore, in order to compare the two distributions accurately, we use a KDE to get $\eta^\pi(x,a)(y_i), \eta^\pi(x,a)(\tilde{y}_i), T^\pi \eta(x',a')(y_i), T^\pi \eta(x',a')(\tilde{y}_i)$.

Let $y_i \sim \eta^\pi(x,a)$ be samples from the predicted return distribution and $\tilde{y}_j \sim T^\pi \eta(x,a)$ be samples from the target distribution. First, we evaluate the KDE of $T^\pi \eta(x,a)$ on the predicted support:

$$T^\pi \eta(x,a)(y_i) = \frac{1}{M} \sum_{j=1}^{M} K_h(y_i - \tilde{y}_j), \quad \forall i = 1, \ldots, N,$$

and reciprocally:

$$\hat{\eta}^\pi(x,a)(\tilde{y}_j) = \frac{1}{N} \sum_{i=1}^{N} K_h(\tilde{y}_j - y_i), \quad \forall j = 1, \ldots, M.$$

The process is illustrated in figure 8 and detailed in algorithm 1.

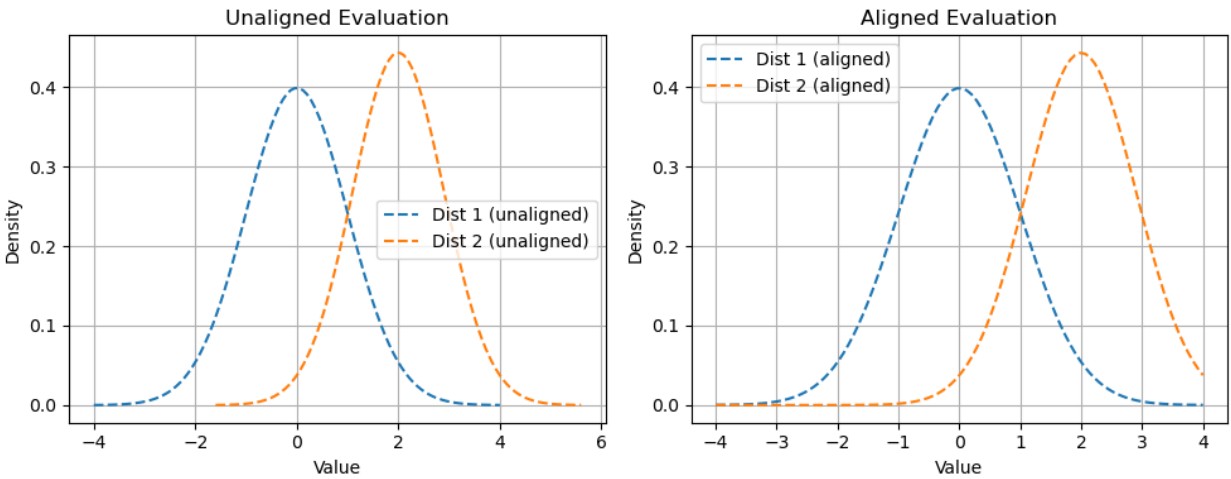

Figure 8: Impact of Support Alignment on Distribution Comparison. Left: Two Gaussian distributions evaluated on different support sets, making direct comparison ill-posed. Right: The same distributions evaluated on a shared support, enabling meaningful density-wise comparison. This illustrates the necessity of support alignment in distributional reinforcement learning, akin to the projection step in C51.

---

**Algorithm 1** Flow-based Target Distribution Construction in Distributional RL

---

**Require:** Current return distribution $\eta^\pi(x, a)$, reward $r$, next state $x'$, terminal indicator $d$, support $z$, number of samples $N$

Compute next-state return distribution:

$$(y', p^\pi(y')) \leftarrow g(F_\theta(x', z), z)$$

**if** $x'$ is terminal **then**

    Replace $(y', p^\pi(y'))$ with Gaussian $\mathcal{N}(r, 0.1)$

**end if**

Compute expected returns:

$$Q^\pi(x', a') = \sum_{y'} y' \cdot p^\pi(y'))$$

Greedy action: $a^* = \arg\max_{a'} Q^\pi(x', a')$

Select corresponding distribution: $y^* = y^*[a^*]$

Select support $s = [-\max(y, y^*), \max(y, y^*)]$

Estimate KDEs:

$$\hat{p}_\eta(s) \leftarrow \text{KDE}(y, \eta(x, a)(y)), \quad \hat{p}_{T^\pi\eta}(s) \leftarrow \text{KDE}(y^*, T^\pi\eta(x', a^*)(y^*))$$

Interpolate both to support $\{y_i\}_{i=1}^N$:

$$p_\eta(y_i), p_{T^\pi\eta}(y_i) \leftarrow \text{Interpolate}(\hat{p}_\eta, \hat{p}_{T^\pi\eta})$$

Interpolate both to support $\{\tilde{y}_j\}_{j=1}^N$:

$$\tilde{p}_\eta(\tilde{y}_j), \tilde{p}_{T^\pi\eta}(\tilde{y}_j) \leftarrow \text{Interpolate}(\hat{p}_\eta, \hat{p}_{T^\pi\eta})$$

**return** $\{\tilde{y}_j\}_{j=1}^N, p_\eta(y_i), p_{T^\pi\eta}(y_i), \tilde{p}_\eta(\tilde{y}_j), \tilde{p}_{T^\pi\eta}(\tilde{y}_j)$

---

### 7.2 Relevant distance properties

Bellemare et al. (2017b) show that the Cramér distance holds the following properties:

**Scale sensitivity**. Consider a divergence $\mathbf{d}$, and for two random variables $X, Y$ with distributions $P, Q$, write $\mathbf{d}(X, Y) := \mathbf{d}(P, Q)$. We say that $\mathbf{d}$ is scale sensitive (of order $\beta$), i.e. it has property (**S**) if there exists a $\beta > 0$ such that for all $X, Y$, and a real value $c > 0$,

$$\mathbf{d}(cX, cY) \leq |c|^\beta \mathbf{d}(X, Y) \tag{S}$$

**Sum invariance**: A divergence $\mathbf{d}$ has property (**I**), i.e. it is sum invariant, if whenever $A$ is independent from $X, Y$

$$\mathbf{d}(A + X, A + Y) \leq \mathbf{d}(X, Y) \tag{I}$$

A divergence is said ideal if it possesses both (**S**) and (**I**).

**Unbiased sample gradients**: Let $X_1, X_2, \ldots, X_m$ be independent samples from $P$ and define the empirical distribution $\hat{P}_m := \hat{P}_m(X_m) := \frac{1}{m}\sum_{i=1}^m \delta_{X_i}$. From this, define the sample loss $\mathbf{d}(\hat{P}_m, Q_\theta)$. We say that $\mathbf{d}$ has unbiased sample gradients when the expected gradient of the sample loss equals the gradient of the true loss for all $P$ and $m$:

$$\mathbb{E}_{X_m \sim P} \nabla_\theta \mathbf{d}(\hat{P}_m, Q_\theta) = \nabla_\theta \mathbf{d}(P, Q_\theta) \tag{U}$$

If a divergence does not possess (**U**), then minimising it with stochastic gradient descent may not converge or towards the wrong minimum. Conversely, if $\mathbf{d}$ possesses (**U**) then we can guarantee that the distribution which minimises the expected sample loss is $Q = P$. From these properties, Bellemare et al. (2017b) draws

the following propositions:

**Proposition 1:** *The KL divergence has unbiased sample gradients (U), but is not scale sensitive (S).*

**Proposition 2:** *The Wasserstein metric is ideal (I, S), but does not have unbiased sample gradients.*

### 7.3 Cramér distance surrogate

We recall the Cramér distance definition. Given two CDFs of a real random variable $X$, denoted as $P(x)$ and $Q(x)$, the Cramér distance is defined as follows:

$$C_p(P,Q) = \left( \int_{-\infty}^{\infty} |P(x) - Q(x)|^p \right)^{1/p}$$

The aim is to approximate this distance using the PDFs of the considered distributions rather that their CDFs. We begin by expressing the CDF difference as an integral over the PDFs:

$$P(x) - Q(x) = \int_{-\infty}^{x} (p(t) - q(t))\, dt \tag{13}$$

Substituting into the Cramér distance:

$$C_2^2(P,Q) = \int_{-\infty}^{\infty} \left( \int_{-\infty}^{x} (p(t) - q(t))dt \right)^2 dx \tag{14}$$

Using Jensen's inequality (or the Cauchy-Schwarz inequality), we obtain an upper bound on the inner square:

$$\left( \int_{-\infty}^{x} (p(t) - q(t))dt \right)^2 \leq |x - y| \int_{-\infty}^{x} (p(t) - q(t))^2 dt \tag{15}$$

for some reference point $y \leq x$. Thus:

$$C_2^2(P,Q) \lesssim \int_{-\infty}^{\infty} \left[ |x - y| \cdot \int_{-\infty}^{x} (p(t) - q(t))^2 dt \right] dx \tag{16}$$

Swapping the integration order (via Fubini's theorem) and discretizing the domain into $N$ evaluation points $\{y_i\}$, we arrive at the following surrogate:

$$\mathcal{L}^2(p,q) = \frac{1}{N^2} \sum_{i=1}^{N} \sum_{j=1}^{N} (p(y_i) - q(y_i))^2 \cdot |y_i - y_j| \tag{17}$$

This expression can be interpreted as a **geometry-weighted $L^2$ distance** between PDFs, where the weights $|y_i - y_j|$ reflect the spatial relationships of the support points and softly approximate the geometry used in optimal transport.

Moreover, it is also structurally related to kernel-based measures like the *energy distance* and *maximum mean discrepancy (MMD)* as it has a pairwise structure similar to energy distance or MMD:

$$\mathbb{E}_{X \sim P, Y \sim Q}[k(X,Y)] - \frac{1}{2}\mathbb{E}_{X,X'}[k(X,X')] - \frac{1}{2}\mathbb{E}_{Y,Y'}[k(Y,Y')]$$

for a geometry-inducing kernel $k(x,y) = |x - y|$.

Therefore, this expression can be viewed as an inner-product kernel on the PDF difference, weighted by a geometry-aware term $|y_i - y_j|$, similar to using a linear kernel with explicit geometric structure.

However this loss, while being an acceptable surrogate for the Cramér distance, does not allow for the operator $T^\pi$ to be a gamma contraction. Therefore we propose another surrogate:

$$\mathcal{L}^2(p, q) = \frac{1}{N^2} \sum_{i=1}^{N} \sum_{j=1}^{N} (p(y_i) - q(y_i))^2 \cdot |y_i - y_j|^2 \tag{18}$$

This surrogate, inspired by the Cramér distance can be thought of as PDF-space surrogate for Cramér-like behaviour. As shown below it holds the same desirable qualities as the Cramér distance (true distance, defines a gamma contraction and has unbiased sample gradients.

In the next sections we will investigate the three following questions:

- **Q1:** Is this loss a proper distance? If so, then the loss function would be symmetric and scale sensitive unlike KL divergence.

- **Q2:** Is the distributional Bellman operator a contraction in this case? If so, this would ensure convergence of the distributional Bellman operator $\eta^\pi$ towards the random returns $T_\eta^\pi$.

- **Q3:** Does it still possess the unbiased sample gradient estimate property? If so, then SGD can be used to optimise this loss function in a RL context unlike the Wasserstein distance. More specifically, we will be able to learn from sample transitions.

### 7.3.1 Q1: Is it a Proper Distance?

We consider the following surrogate loss:

$$d(\pi_1, \pi_2) = \frac{1}{N^2} \left( \sum_{i,j} (\pi_1(y_i) - \pi_2(y_i))^2 \cdot |y_i - y_j|^2 \right)^{1/2}, \tag{19}$$

where $\pi_1$ and $\pi_2$ are two probability density functions evaluated over a common finite support $\{y_1, \ldots, y_N\} \subset \mathbb{R}$. We examine whether this function defines a proper distance.

**Non-negativity.** Each term in the sum is non-negative, so $d(\pi_1, \pi_2) \geq 0$.

**Symmetry.** The squared difference $(\pi_1(y_i) - \pi_2(y_i))^2$ is symmetric in $\pi_1$ and $\pi_2$, as is the weight $|y_i - y_j|^2$. Therefore, $d(\pi_1, \pi_2) = d(\pi_2, \pi_1)$.

**Identity of indiscernibles.** Suppose $d(\pi_1, \pi_2) = 0$. Then for all $i$, we must have $\pi_1(y_i) = \pi_2(y_i)$, since all weights $|y_i - y_j|^2$ are positive. Hence, $\pi_1 \equiv \pi_2$ over the support.

**Triangle Inequality.** Let $\delta_i = \pi_1(y_i) - \pi_2(y_i)$. The expression for $d(\pi_1, \pi_2)$ can be rewritten by factoring out $\delta_i^2$ from the inner summation over $j$:

$$d(\pi_1, \pi_2) = \frac{1}{N^2} \left( \sum_{i=1}^{N} (\pi_1(y_i) - \pi_2(y_i))^2 \sum_{j=1}^{N} |y_i - y_j|^2 \right)^{1/2}$$

Let $W_i = \sum_{j=1}^{N} |y_i - y_j|^2$, then:

$$d(\pi_1, \pi_2) = \frac{1}{N^2} \left( \sum_{i=1}^{N} (\pi_1(y_i) - \pi_2(y_i))^2 W_i \right)^{1/2}$$

Consider a vector $\mathbf{a} \in \mathbb{R}^N$ whose $i$-th component is $a_i = (\pi_1(y_i) - \pi_2(y_i))\sqrt{W_i}$. Then, the sum inside the square root is the squared Euclidean ($L_2$) norm of $\mathbf{a}$:

$$\sum_{i=1}^{N} (\pi_1(y_i) - \pi_2(y_i))^2 W_i = \sum_{i=1}^{N} a_i^2 = \|\mathbf{a}\|_2^2$$

Thus, the distance can be written as:

$$d(\pi_1, \pi_2) = \frac{1}{N^2} \|\mathbf{a}\|_2$$

For any distribution $\pi_k$, define a vector $\mathbf{v}_k \in \mathbb{R}^N$ whose $i$-th component is $\pi_k(y_i)\sqrt{W_i}$:

$$\mathbf{v}_k = \begin{pmatrix} \pi_k(y_1)\sqrt{W_1} \\ \pi_k(y_2)\sqrt{W_2} \\ \vdots \\ \pi_k(y_N)\sqrt{W_N} \end{pmatrix}$$

Then, the vector $\mathbf{a}$ is simply the difference between $\mathbf{v}_1$ and $\mathbf{v}_2$:

$$\mathbf{a} = \mathbf{v}_1 - \mathbf{v}_2$$

Substituting this into the expression for $d(\pi_1, \pi_2)$:

$$d(\pi_1, \pi_2) = \frac{1}{N^2} \|\mathbf{v}_1 - \mathbf{v}_2\|_2$$

To prove the triangle inequality for $d$, we need to show that $d(\pi_1, \pi_3) \le d(\pi_1, \pi_2) + d(\pi_2, \pi_3)$. Substitute the expression from above:

$$\frac{1}{N^2} \|\mathbf{v}_1 - \mathbf{v}_3\|_2 \le \frac{1}{N^2} \|\mathbf{v}_1 - \mathbf{v}_2\|_2 + \frac{1}{N^2} \|\mathbf{v}_2 - \mathbf{v}_3\|_2$$

Multiplying both sides by $N^2$:

$$\|\mathbf{v}_1 - \mathbf{v}_3\|_2 \le \|\mathbf{v}_1 - \mathbf{v}_2\|_2 + \|\mathbf{v}_2 - \mathbf{v}_3\|_2$$

This last inequality is the fundamental triangle inequality property of the Euclidean ($L_2$) norm in $\mathbb{R}^N$. Since the Euclidean norm is a valid norm, this inequality holds true for any vectors $\mathbf{v}_1, \mathbf{v}_2, \mathbf{v}_3 \in \mathbb{R}^N$.

Therefore, the given loss function $d(\pi_1, \pi_2)$ satisfies the triangle inequality.

### 7.3.2 Q2: Is the Distributional Bellman Operator a Contraction for d?

Before proving the sought result, we first need to state the following lemma:

Let $\pi_1$ and $\pi_2$ be two probability density functions **evaluated** over the support $\{y_1, \ldots, y_N\} \subset \mathbb{R}$. For any scalar $a \in \mathbb{R}$, define the scaled PDFs $\pi_1(a)$ and $\pi_2(a)$ over the support $\{ay_1, \ldots, ay_N\}$, such that:

$$\pi_k^{(a)}(ay_i) = \frac{1}{|a|} \pi_k(y_i), \quad \text{for } k = 1, 2.$$

Then, the metric **d** is invariant under scaling of the input space.

$$d(\pi_1^{(a)}, \pi_2^{(a)}) = d(\pi_1, \pi_2).$$

Similarly, for a shift $b \in \mathbb{R}$, define the translated PDFs $b_{r,\cdot}\#\pi_1$ and $b_{r,\cdot}\#\pi_2$. Then the metric satisfies the translation invariance:

$$d(b_{r,\cdot}\#\pi_1, b_{r,\cdot}\#\pi_2) = d(\pi_1, \pi_2)$$

*Proofs:*

**Scaling.** Let $a \in \mathbb{R}$ be a scalar. Define the scaled densities as $\pi_k^{(a)}(ay_i) = \frac{1}{|a|}\pi_k(y_i)$ for $k = 1, 2$. Then for all $a \in \mathbb{R}^*$:

$$
\begin{aligned}
d(\pi_1^{(a)}, \pi_2^{(a)}) &= \left( \frac{1}{N^2} \sum_{i,j} \left( \pi_1^{(a)}(ay_i) - \pi_2^{(a)}(ay_i) \right)^2 \cdot |ay_i - ay_j|^2 \right)^{1/2} \\
&= \left( \frac{1}{N^2} \sum_{i,j} \left( \frac{1}{|a|}(\pi_1(y_i) - \pi_2(y_i)) \right)^2 \cdot |a|^2 \cdot |y_i - y_j|^2 \right)^{1/2} \\
&= \left( \frac{1}{N^2} \sum_{i,j} \frac{1}{|a|^2}(\pi_1(y_i) - \pi_2(y_i))^2 \cdot |a|^2 \cdot |y_i - y_j|^2 \right)^{1/2} \\
&= \left( \frac{1}{N^2} \sum_{i,j} \left( (\pi_1(y_i) - \pi_2(y_i))^2 \cdot |y_i - y_j|^2 \right) \right)^{1/2} \\
&= d(\pi_1, \pi_2).
\end{aligned}
\tag{20}
$$

Therefore, the metric $d$ is invariant under scaling of the input space.

**Translation Invariance.** Let $b \in \mathbb{R}$ be a translation constant. Define the translated densities as:

$$
(T_b\pi)(x) := \pi(x - b)
\tag{21}
$$

We aim to prove:

$$
d(T_b\pi_1, T_b\pi_2) = d(\pi_1, \pi_2)
\tag{22}
$$

**Proof**

Consider the squared distance:

$$
d^2(T_b\pi_1, T_b\pi_2) = \int_{\mathbb{R}} \int_{\mathbb{R}} (T_b\pi_1(y) - T_b\pi_2(y))^2 \cdot |y - z|^2 \, dy \, dz
\tag{23}
$$

$$
= \int_{\mathbb{R}} \int_{\mathbb{R}} (\pi_1(y - b) - \pi_2(y - b))^2 \cdot |y - z|^2 \, dy \, dz
\tag{24}
$$

Apply the change of variables: $u = y - b$, $v = z - b$, so $dy = du$, $dz = dv$, and $|y - z| = |u - v|$. Then:

$$
d^2(T_b\pi_1, T_b\pi_2) = \int_{\mathbb{R}} \int_{\mathbb{R}} (\pi_1(u) - \pi_2(u))^2 \cdot |u - v|^2 \, du \, dv
\tag{25}
$$

$$
= d^2(\pi_1, \pi_2)
\tag{26}
$$

Thus, the surrogate distance is translation invariant.

**Discrete Case Justification**

In the discrete implementation, the surrogate is approximated over a grid $\{y_i\}_{i=1}^N$ as:

$$
d_{\text{disc}}(\pi_1, \pi_2) = \left( \frac{1}{N^2} \sum_{i,j} (\pi_1(y_i) - \pi_2(y_i))^2 \cdot |y_i - y_j|^2 \right)^{1/2}
\tag{27}
$$

Suppose both densities are translated by $b$, and the support points are also translated: $y_i \mapsto y_i + b$. Then:

$$d_{\text{disc}}(\pi_1^{(+b)}, \pi_2^{(+b)}) = \left( \frac{1}{N^2} \sum_{i,j} (\pi_1(y_i) - \pi_2(y_i))^2 \cdot |(y_i + b) - (y_j + b)|^2 \right)^{1/2} \tag{28}$$

$$= \left( \frac{1}{N^2} \sum_{i,j} (\pi_1(y_i) - \pi_2(y_i))^2 \cdot |y_i - y_j|^2 \right)^{1/2} \tag{29}$$

$$= d_{\text{disc}}(\pi_1, \pi_2) \tag{30}$$

In Bellemare et al. (2023), the authors show that the distributional Bellman operator $T^\pi$ is a $\gamma$-contraction under the metric

$$\bar{d}_p(\eta_1, \eta_2) = \sup_{s,a} d_p\left(\eta_1(s,a), \eta_2(s,a)\right),$$

where $d_p$ denotes the $p$-Wasserstein distance, and $\eta_1, \eta_2$ are return distributions over state-action pairs.

Since our proposed metric $d$ satisfies the same two key properties—positive homogeneity and translation invariance—the same proof structure applies. For completeness, we reproduce the argument using $d$ instead of $d_p$. By definition we have:

$$\bar{d}(T^\pi \eta_1, T^\pi \eta_2) = \sup_{x,a} d\left(T^\pi \eta_1(x,a), T^\pi \eta_2(x,a)\right) \tag{31}$$

By the properties of $d$, we have for any state-action pair $(x, a)$:

$$
\begin{aligned}
d\left(T^\pi \eta_1(x,a), T^\pi \eta_2(x,a)\right) &= d\left(b_{R(x,a),\gamma} \# \eta_1^\pi(x', a'), \ b_{R(x,a),\gamma} \# \eta_2^\pi(x', a')\right) \\
&= \gamma\, d\left(\eta_1(x', a'), \ \eta_2(x', a')\right) \quad \text{by scaling and translation invariance} \\
&\leq \gamma \sup_{x',a'} d\left(\eta_1(x', a'), \eta_2(x', a')\right)
\end{aligned}
\tag{32}
$$

Taking the supremum over all $(x, a)$, we obtain:

$$\bar{d}(T^\pi \eta_1, T^\pi \eta_2) \leq \gamma \sup_{x',a'} d(\eta_1(x', a'), \eta_2(x', a')) = \gamma \bar{d}(\eta_1, \eta_2) \tag{33}$$

Hence, $T^\pi$ is a $\gamma$-contraction under $\bar{d}$. By Banach's fixed-point theorem, $T^\pi$ admits a unique fixed point $\eta^\pi$. Assuming all moments are bounded, the sequence $(\eta_k)$ generated by iterated application of $T^\pi$ converges to $\eta^\pi$ in $\bar{d}$.

### 7.3.3 Q3: Does d Admit Unbiased Sample Gradient Estimates?

In Bellemare et al. (2017a), the authors highlight a key limitation of the Wasserstein loss in reinforcement learning: although it has appealing theoretical properties such as contraction, it cannot be minimized using unbiased stochastic gradients. This stems from the fact that in practice, learning proceeds from sample transitions, while the Wasserstein loss is not compatible with such sample-based optimization.

More precisely, define the Wasserstein sample loss between a return distributions $\eta^\pi$ and its projected target $(b_{R(x,a),\gamma})\#\eta(x')$

$$\mathcal{L}_W(\theta) := d_p\left(\eta^\pi(x,a), (b_{R(x,a),\gamma})\#\eta(x')\right),$$

where $x' \sim \mathcal{P}(\cdot|x,a)$.

As shown in Bellemare et al. (2017a), this sample loss provides an upper bound on the true Wasserstein loss:

$$\mathcal{L}_W(\theta) \leq \mathbb{E}_{R,x'}\hat{\mathcal{L}}_W(\theta; R, x'),$$

with strict inequality in general. Consequently, minimizing the sample loss does not guarantee convergence to the true minimizer of the population loss.

In Bellemare et al. (2017b), this phenomenon is formalized through the notion of *unbiased sample gradients*, denoted property (**U**). They prove that the Wasserstein distance does not satisfy (**U**), meaning that its sample gradients are not unbiased estimators of the gradient of the expected loss.

As an example, consider $P = \mathcal{B}(\theta^*)$ and $Q_\theta = \mathcal{B}(\theta)$, two Bernoulli distributions. Suppose we draw $m$ samples from $P$ to obtain the empirical distribution $\hat{P}_m$. Then, the following inequality generally holds:

$$\arg\min_\theta \mathbb{E}\left[W_p(\hat{P}_m, Q_\theta)\right] \neq \arg\min_\theta W_p(P, Q_\theta).$$

Thus, stochastic optimization with the Wasserstein distance can converge to incorrect solutions, even with large samples.

In contrast, we will prove that our distance $d$ satisfies property (**U**). Specifically, we will show:

$$\arg\min_\theta \mathcal{L}(\theta) = \arg\min_\theta \mathbb{E}_{R,x'}\mathcal{L}(\theta, R, x')$$

*Proof:* Consider a random variable $X$ and $X_N := \{X_1, \ldots, X_N\}$ drawn from a distribution $P$, the empirical distribution $\hat{P}_N := \frac{1}{N}\sum_{i=1}^N \delta_{X_i}$, and a distribution $Q_\theta$. We have:

$$d(\hat{P}_N, Q_\theta) = \frac{1}{N^2}\left(\sum_{i,j}\left(\hat{P}_N(X_i) - Q_\theta(X_i)\right)^2 \cdot |X_i - X_j|^2\right)^{1/2}, \tag{34}$$

We denote:

$$f(\hat{P}_N, Q_\theta) := \sum_{i,j=1}^N (\hat{P}_N(X_i) - Q_\theta(X_i))^2 \cdot |X_i - X_j|^2 \tag{35}$$

Then:

$$d(\hat{P}_N, Q_\theta) = \left(\frac{1}{N^2}f(\hat{P}_N, Q_\theta)\right)^{1/2} \tag{36}$$

Therefore:

$$\nabla_\theta d(\hat{P}_N, Q_\theta) = \frac{1}{2N\sqrt{f}} \cdot \nabla_\theta f \tag{37}$$

And:

$$\nabla_\theta f = \sum_{i,j} 2(\hat{P}_N(X_i) - Q_\theta(X_i)) \cdot (-\nabla_\theta Q_\theta(X_i)) \cdot |X_i - X_j|^2 \tag{38}$$

Note that:

1. $\hat{P}_N(X_i) = \frac{1}{N}\sum_{k=1}^N \mathbf{1}_{\{X_k = X_i\}}$ is an unbiased estimator of $P(X_i)$

2. $\nabla_\theta f$ depends linearly on $\hat{P}_N(X_i)$, and $\mathbb{E}[\hat{P}_N(X_i)] = P(X_i)$.

Therefore:

$$\mathbb{E}[\nabla_\theta f(\hat{P}_N, Q_\theta)] = \nabla_\theta f(P, Q_\theta) \tag{39}$$

Thus:

$$\mathbb{E}[\nabla_\theta d(\hat{P}_m, Q_\theta)] = \nabla_\theta d(P, Q_\theta) \tag{40}$$

**Example.** Below is an illustration using the Bernoulli distribution.

Consider the simple MDP shown in Figure 9, where taking action $A$ leads to two possible outcomes with equal probability, yielding rewards 0 and 1 respectively.

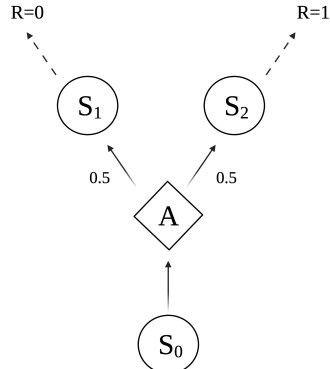

Figure 9: Example of stochastic MDP, where the same action $A$ leads to 2 different states with equal probability, and different rewards $R$.

Consider the distribution $P$ defined as:

$$P = \begin{cases} 0 & \text{w.p} \quad 1/2 \\ 1 & \text{w.p} \quad 1/2 \end{cases} \tag{41}$$

We consider a model distribution $Q$, parameterized by $p$:

$$Q = \begin{cases} 0 & \text{w.p} \quad p \\ 1 & \text{w.p} \quad 1-p \end{cases} \tag{42}$$

The distance between $P$ and $Q$ is

$$\begin{aligned} d(P,Q) &= (p-1/2)^2 + (1-p-1/2)^2 \\ &= 2p^2 - 2p + 1/2 \end{aligned} \tag{43}$$

There $d = 0 \iff p = 1/2$.

Suppose now we observe a single sample. If we observe 0, the empirical distribution $\hat{P}$ places all mass on 0, and

$$d(\hat{P}, Q) = (\hat{P}(0) - Q(0))^2 = (1-p)^2$$

If we observe 1, we get

$$d(\hat{P}, Q) = p^2$$

Thus, the expected sample loss is

$$\begin{aligned} \mathbb{E}\left[d(\hat{P}, Q)\right] &= \frac{1}{2}(1-p)^2 + \frac{1}{2}p^2 \\ &= \frac{1}{2} + p^2 - p \end{aligned} \tag{44}$$

Taking the derivative:

$$\begin{aligned} \nabla \mathbb{E}\left[d(\hat{P}, Q)\right] &= 0 \\ \iff p &= \frac{1}{2} \end{aligned} \tag{45}$$

This example shows that although $\mathbb{E}\left[d(\hat{P}, Q)\right] > d(P, Q)$, both losses are minimized at the same parameter $p$:

$$\arg\min_p d(P, Q) = \arg\min_p \mathbb{E}[d(\hat{P}, Q)] \qquad \text{and} \qquad \nabla d(P, Q) = \nabla\mathbb{E}[d(\hat{P}, Q)]$$

Both distances reach their minimum for the same parameter $p$, and optimising one or the other through SGD should converge towards the same parameters. Next we will prove this result more generally, although the proof is similar.

**Arbitrary Bernoulli distribution.** Consider again $P = \mathcal{B}(\theta^*)$ and $Q_\theta = \mathcal{B}(\theta)$. The empirical distribution $\hat{P}_m$ is characterized by $\hat{\theta} = \frac{1}{m}\sum_{i=1}^{n} \mathbb{I}_A$.

The true loss and the derivative are:

$$\begin{aligned}
d(P, Q) &= (\theta^* - \theta)^2 + (1 - \theta^* + \theta - 1)^2 \\
&= 2(\theta - \theta^*)^2 \\
\nabla_\theta d(P, Q) &= 4(\theta - \theta^*)
\end{aligned} \tag{46}$$

Similarly, the sample loss and its gadient:

$$\begin{aligned}
d(\hat{P}, Q) &= (\hat{\theta} - \theta)^2 + (1 - \hat{\theta} + \theta - 1)^2 \\
&= 4(\theta - \hat{\theta})^2 \\
\nabla_\theta d(\hat{P}, Q) &= 4(\theta - \hat{\theta})
\end{aligned} \tag{47}$$

Taking expectation over samples:

$$\begin{aligned}
\mathbb{E}_m \nabla_\theta d(\hat{P}, Q) &= 4\mathbb{E}_m(\theta - \hat{\theta}) \\
&= 4\mathbb{E}_m(\theta - \frac{1}{m}\sum \mathbb{I}_A) \\
\lim_{m \to \infty} \mathbb{E}_m \nabla_\theta d(\hat{P}, Q) &= 4(\theta - \theta^*)
\end{aligned} \tag{48}$$

Where in the penultimate line, we used the fact that $\frac{1}{m}\sum \mathbb{I}_A$ is an unbiased estimator of $\theta^*$. Therefore:

$$\lim_{m \to \infty} \mathbb{E}_m \nabla_\theta d(\hat{P}, Q) = \nabla_\theta d(P, Q)$$

Finally,

$$\arg\min_\theta \mathbb{E}[d(\hat{P}, Q)] = \arg\min_\theta \nabla_\theta d(P, Q)$$

To conclude, we proved that:

- Our loss function is a proper distance

- The distributional Bellman operator is a $\gamma$-contraction in $\bar{d}$

- The minimum of the expected sample loss is the same as the minimum of the true loss.

### 7.4 Toy Markov decision processes

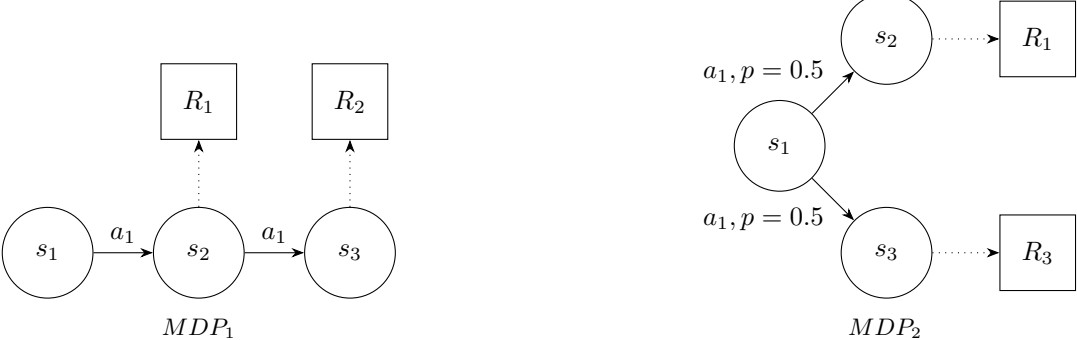

Figure 10: Two example MDPs.

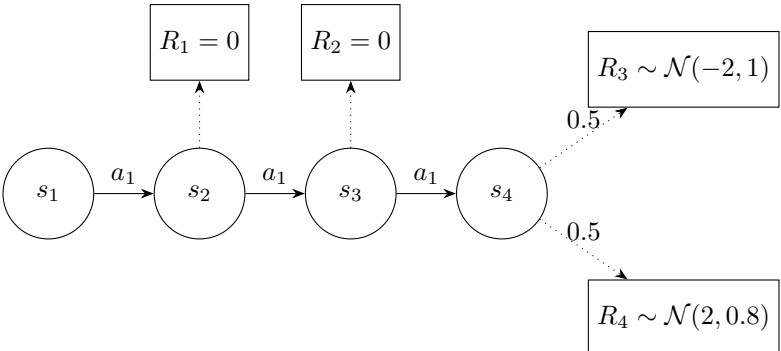

Figure 11: $MDP_3; \gamma = 1$

### 7.5 Additional results

#### 7.5.1 Frozen Lake

Figure 13 illustrates the Frozen Lake environment, a stochastic grid-world designed to highlight multimodal value distributions. An agent must navigate from a starting point to a goal while avoiding holes in the ice. Due to slipperiness, intended moves may result in perpendicular actions; for example, choosing to move right results in a 1/3 chance of moving right, up, or down. This high level of stochasticity induces significant variability in returns, making Frozen Lake a suitable testbed for visualizing multimodal Q-value distributions. The learned value distributions for each state-action pair, obtained using the surrogate loss, are displayed in Figure 12. States are ordered left to right, top to bottom.

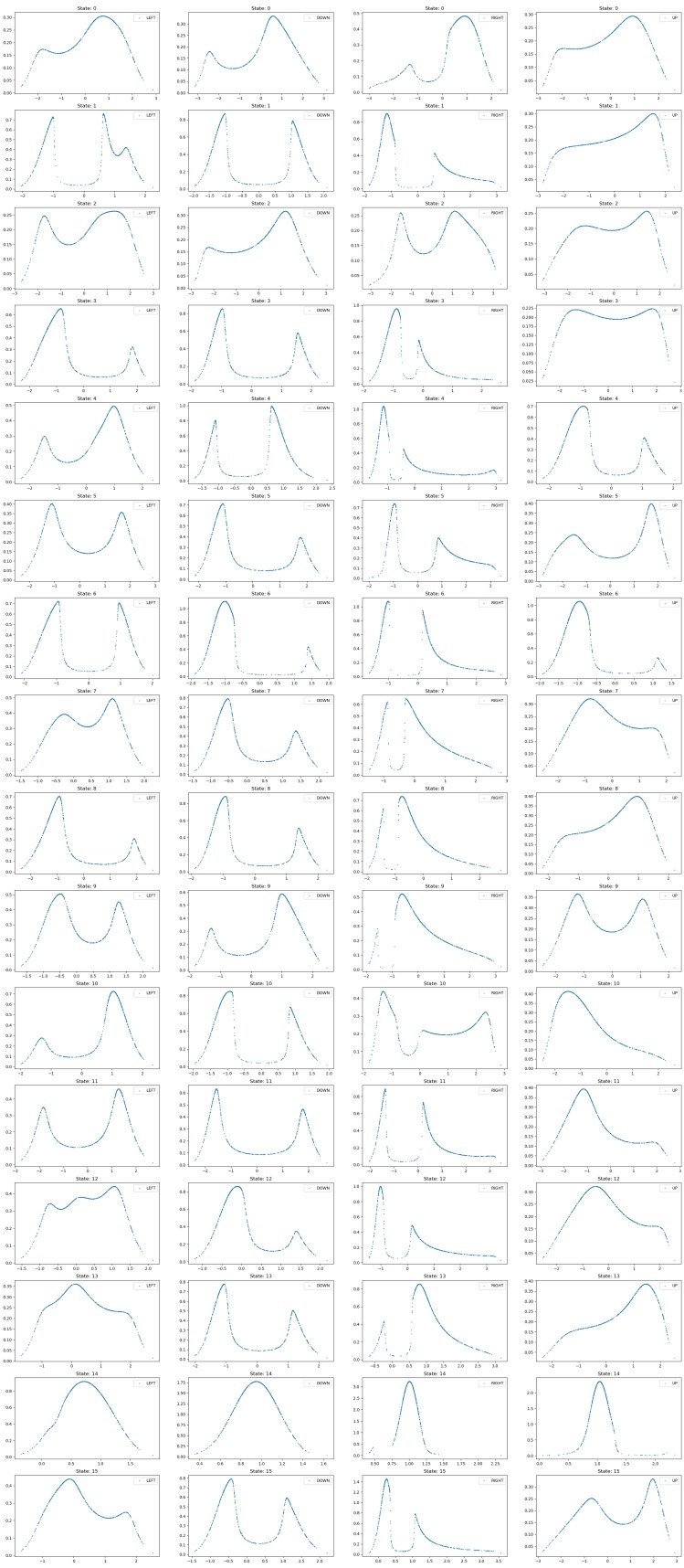

Figure 12: Learnt value distributions using the surrogate loss for each state-action pair of the Fozen Lake environment. States are numbered from left to right and up to down, i.e upper left is 0, bottom right is 16.

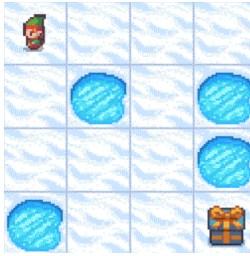

Figure 13: Frozen lake environment

### 7.5.2 ATARI-5 Raw scores

Table 2: Raw scores on ATARI-5 Benchmark.

| Games | Random | Human | DQN | C51 | IQN | NFDRL-E | NFDRL-S |
|---|---|---|---|---|---|---|---|
| Battle Zone | 2,360.0 | 37,187.5 | 29,900.0 | 28742.0 | **42244.0** | 32800.0 | **34540.0** |
| Double Dunk | -18.6 | -16.4 | -6.6 | 2.5 | 5.6 | **7.8** | 7.6 |
| Name this Game | 2,292.3 | 8,049.0 | 8,207.8 | 12,542.0 | **22,682.0** | 15,667.2 | **17,309.5** |
| Phoenix | 761.4 | 7,242.6 | 8,485.2 | 17,490 | **56,599** | 18,914 | **20,042** |
| Q*Bert | 163.9 | 13,455.0 | 13,117.3 | 23,784 | 25,750 | 22,671 | **25,852** |

### 7.6 Implemetation Details and hyperparameters

**Architecture.** We base all baselines and our method on the same underlying neural network. Its architecture and training loop composition follows the structure used in CleanRL (Huang et al., 2022). Our method however requires 4 separate heads outputing $\{w_i^j, \mu_i^j, \sigma_i^j\}$ and $G_{\max}^j$. Hyperparameters are displayed in table 3.

As mentioned in section 3.4, we choose $\sigma = 0.05$ to ensure the resulting Gaussian is sharply peaked around $r$, while still numerically stable. This value is derived by matching the resolution of the C51 model, which discretizes the support $[0, 10]$ into 51 bins, yielding a bin width of approximately 0.2. To ensure that 95% of the Gaussian's mass lies within the central bin (i.e., $[r - 0.1, r + 0.1]$), we solve $2\sigma = 0.1$, yielding $\sigma = 0.05$. Of course, different values can be chosen to allow for sharper distributions.

**Action selection.** During training, actions are chosen using an $\epsilon$-greedy strategy: with probability $\epsilon$, a random action is selected; otherwise, the model selects the best action. The value of $\epsilon$ decreases over time. In the latter case, action selection follows the approach of C51 or IQN. The model includes one head per possible action, each outputting the parameters of a Gaussian mixture and $G^{max}$—defining a flow function per action. From these, $n$ samples are drawn from the base distribution and passed through the corresponding flow. This yields $n$ return values and densities per action. The expected return is computed for each, and the action with the highest expected value is chosen.

| Hyperparameter | Value / Description |
|---|---|
| total_timesteps | 10,000,000 (total timesteps of the experiments) |
| learning_rate | 5e-5 (learning rate of the optimizer) |
| max_norm | 3.0 (maximum allowable gradient norm) |
| num_envs | 4 (number of parallel game environments) |
| buffer_size | 1,000,000 (size of the replay memory buffer) |
| gamma | 0.99 (discount factor) |
| target_network_frequency | 1 (steps between target network updates) |
| batch_size | 64 (batch size from replay memory) |
| start_e | 1.0 (starting epsilon for exploration) |
| end_e | 0.01 (final epsilon for exploration) |
| exploration_fraction | 0.2 (fraction of timesteps for epsilon decay) |
| learning_starts | 30,000 (timestep to start learning) |
| train_frequency | 4 (training frequency) |
| hidden_size_1 | 512 (size of first hidden layer) |
| hidden_size_2 | 256 (size of second hidden layer) |
| n_flows | 1 (number of flows) |
| n_components | 4 (number of components in the mixture) |
| n_samples | 500 (samples drawn from the base distribution) |
| final_reward_variance | 0.1 (final state reward normal distribution variance) |
| bandwidth | 0.05 (KDE bandwidth) |

Table 3: List of hyperparameters used in our experiments.

