# OpenReview forum: "Flow Models for Unbounded and Geometry-Aware Distributional Reinforcement Learning"
_TMLR — Rejected by TMLR_

### Review · Reviewer_XicL · 2025-05-27

**Summary Of Contributions:**

This paper proposes to use Normalized Flows as a generative technique to better model returns distribution in the setting of distributional reinforcement learning. The idea is to use NF to move from a base distribution to that of the agent's accumulated return, in a way that is invertible, avoids discretization and is easy to sample from and learn. Empirical results show that modelling a CDF is helpful in improving the representational capacity of the proposed algorithm in some peculiar settings, while at the same time not deteriorating too much performances in more complex scenarios.

**Audience:**

Yes

**Claims And Evidence:**

Yes

**Requested Changes:**

- The symbol \# used to define the pushforward function is never actually explained, and so it is not the meaning of pushforward function itself. The background section on both DistRL and NF is a bit too concise, and is hard for an unfamiliar reader to get the gist of how these things works from that alone.

- Why cannot we observe true returns? This is true only in the infinite-horizon case, but for episodic RL settings we can indeed sample complete returns by interacting with the environment (e.g., the Monte-Carlo approximation of the $Q$-function done in REINFORCE). In the following (Section 3.4) you mention final states, which lead me to think that you are discussing episodic settings here (even though the summations over time-steps in the definition of $V^{\pi}(s)$ and $Q^{\pi}(s,a)$ were infinite). I would suggest you better clarify this aspect, and eventually correct the previous statement.

- As a non-expert in DistRL, it is not entirely clear to me how you sample the action $a^j$ to perform in the environment from the output of the network $h_{\theta}$. This may be a trivial thing for people more used to this setting than me, but I think it may be a good to make everything clear and as straightforward as possible.

- What are $\hat{\eta}$ and $\hat{T}^{\pi}\eta$ in the loss expression on page 6? How are these defined?

- More in general, you claim that one of the benefits of using CDF flows over other flow typologies is that they do not require discretization, but then you employ a methodology similar to that of C51, which is indeed based on discretization. Are you not losing the benefit of CDFs here? Could you discuss this aspect a bit more in depth?

- I think you are missing the differential $dx$ in Equation (9)?

- Results in Figure 2(f) are a bit difficult to interpret: it seems like your surrogate Cramèr loss is producing a wider distribution when the bandwidth becomes smaller, as opposed as what the true Cramèr loss is doing (and what intuition would suggest). Do you have any insight on why this is the case?

- More generally, labelling the plot axes is a good practice, and help the reader in understanding them. I guess the y-axis here (and the next figures) is the number of samples?

- "This flexibility contrasts with existing methods: (1) C51 tends to learn broader distributions when rewards fall between fixed atoms; (2) Quantile-based methods (QR-DQN, IQN) represent distributions as sums of Dirac masses at fixed quantiles, lacking control over local spread." Is this insight plotted somewhere? Or at least in any reference provided? Because otherwise it is difficult to assess the validity of such a claim in practice, with nothing to show it holds...

- "Notably, NFDRL-S, which uses a surrogate Cramèr loss, slightly outperforms the exact version on average, highlighting its favourable trade-off between computational efficiency and performance." The trade-off between computational efficiency and performance imposed by your surrogate Cramèr loss should make the method more efficient (using an approximation rather than the real value) but *less* performing. However, results goes in the other direction (better performance). The motivation you are giving is not convincing to me, and does not explain what is happening in practice. Could you discuss this in a bit more detail?

**Strengths And Weaknesses:**

I generally like the idea of using a technique from the generative AI field to model the return distributions, which is (up to my knowledge, which is limited in the field of distributional RL) fairly new. It seems like a promising direction to improve the learning capacities of the RL agent, and in general to be able to achieve arbitrarily complex results (given enough computing power). However, I think that the paper presentation is in general not strong enough, and it often happens that, after reading a paragraph or a section, you have some questions in mind. Please see the more detailed points below for further explanations. Also, while I appreciate the quality of the empirical results on the simpler MDPs, the results on the Atari environments are not really close to that of the IQN baseline, which basically always outperform your proposed algorithm by quite a margin. I know that it is not always possible to achieve SOTA performances everywhere, but these results left me with the question of why should I prefer your methodology instead of IQN itself, at least in complex enough settings. I think that a bit more of discussion in trying to highlight the benefits of your solution w.r.t. this baseline would greatly help in making it look more compelling even with those inferior results.

---

> ### Author Response · Authors · 2025-06-17
> **Answer to reviewer XicL**
>
> Thank you for your insightful remarks.
>
> **Action choice:** A more detailed implementation details section has been added to the appendix (section 7.6).
>
> **Meaning of** $\mathbf{\hat{\eta}}$. We added a definition in section 3.4.
>
> **Discretization.** A key motivation for using CDF-based flows is to move beyond fixed discretizations like those in C51. While our method samples a finite number of points to compute the loss, as commonly done in continuous distribution modeling, this does not imply discretizing the return distribution itself.
> Importantly, unlike C51:
> - The support of our predicted distribution is not fixed but learned and adapted via the $G^{max}$ transformation, allowing for flexible, unbounded support.
> - The evaluation points $y_i$ used in the surrogate Cramér loss are not atoms defining the distribution, but integration points for approximating the distance between densities.
>
> Thus, our method preserves the main benefit of flows: continuous modeling with adaptive support, without imposing rigid discretization through the loss.
>
> **Surrogate wider distribution.** We believe this is due to the fact that PDFs only reflect local density at a point, not the cumulative mass up to that point. If the predicted and target distributions are close but slightly shifted, their PDFs may not overlap at all. In this case, the surrogate sees a large difference but gives no direction for aligning them (the gradient vanishes or is uninformative). In contrast, CDFs accumulate mass and will differ everywhere after a mismatch, giving strong, global gradients to correct misalignment.
>
> **Plot labelling.** Figures 2, 3 and 4 represent return distributions. The x axis is return values while the y axis is their corresponding density computed using the change of variable formula. We added this description to their captions.
>
> **C51 broader learnt distributions.** We have no reference that explicitly displays this but it is a known and logical behaviour of 51. Consider a situation where the distribution is represented with bins where each bin spans an interval of length 1. Said otherwise, the categorical representation allocates bin 1 to returns between 0 and 1, bin 2 to returns between 1 and 2. Now consider that the return to predict is exactly 1. This value falls right between bin 0 and bin 1. C51 will allocate the same mass to both bins, effectively displaying a wider distribution than it should.
>
> **QR-DQN and IQN lack control over local spread.** QR-DQN predicts a fixed set of return quantiles (e.g., 5th, 15th, …, 95th percentiles), while IQN learns a function to estimate quantiles for arbitrary probabilities sampled from a uniform distribution. Although IQN offers more flexibility in querying quantiles, both methods ultimately represent the return distribution as a discrete set of quantile points.
>
> These predicted quantiles are best interpreted as Dirac masses, each carrying equal probability mass, forming a discrete approximation of the distribution. This representation lacks information about the distribution's behavior between quantile points. As a result, quantile-based methods cannot distinguish between smooth distributions and those with sharp peaks or flat regions between quantiles. They also lack the ability to explicitly model local density or control how probability mass is spread across intervals.
>
> In contrast, our flow-based approach models a continuous PDF, allowing explicit control over the allocation of probability mass across the return space. This enables the model to capture fine-grained structure and local variability that quantile-based methods cannot.
>
> **NFDRL-S performance.** Although NFDRL-S uses a surrogate Cramér loss, its slightly better performance compared to the exact version is consistent with trends in deep learning. The exact Cramér loss introduces sampling noise, discretization errors, and sensitivity to distribution tails, which can destabilize training.
>
> In contrast, the surrogate loss may better handle these challenges, implicitly reweighting or smoothing the distribution, making it more amenable to optimization. This reflects a broader pattern in deep learning where approximations can lead to better generalization and more stable training, as seen in areas like computer vision with perceptual losses.
>
> **Why cannot we observe true returns?** Our statement refers specifically to the setting of off-policy, value-based reinforcement learning, such as DQN-style algorithms, which our method builds upon. In this context, return samples are not directly observed but are approximated via bootstrapping. While Monte Carlo estimates of returns can be computed in on-policy, episodic methods like REINFORCE, these approaches require full trajectories collected under the current policy. Our approach instead leverages experience replay and off-policy updates, where such full return trajectories are generally unavailable. Therefore, the use of bootstrapped targets remains necessary in our case.

---

> > ### Comment · Reviewer_XicL · 2025-06-18
> > **Reply to the Authors**
> >
> > I would like to thank the authors for their useful rebuttal, which addressed most of my concerns. Below are some additional points of discussion:
> >
> > **Surrogate wider distribution**: I find this a reasonable explanation. However, given that its conclusions goes against the immediate intuition of how things should work, I think that it should be explicitly stated in the paper, in order to avoid confusion or counter-intuitive reasoning.
> >
> > **C51 broader learnt distributions** and **QR-DQN and IQN lack control over local spread**: even though I find the explanation quite valid, in the absence of a proper reference that shows these phenomenons clearly, I still find it difficult to trust blindly such a bold claim, from a reader perspective. You either find references that you can cite that supports this, or you have to motivate these claims yourselves. Indeed, it took me your additional explanation to grasp these, and I still have to believe you on these behaviours really happening in practice, for as obvious as they may look now.
> >
> > **NFDRL-S performance**: again, I can buy your explanation here, but it needs to be added to the paper in order to avoid any confusion.

---

> > > ### Author Response · Authors · 2025-06-18
> > > **Reply to reviewer XiCL**
> > >
> > > We thank the reviewer for his appraisal and for his help for making the paper stronger.
> > >
> > > **Surrogate wider distribution**. We added this explanation to our comment on the results for MDP1
> > >
> > > **C51 broader learnt distributions** and **QR-DQN and IQN lack control over local spread**. We understand the need to prove our claim and thank the reviewer for giving us the opportunity to do so. We added a short notebook in the supplementary material that shows both effects. The first part implements a simple situation showing that C51 assigns by design wider distributions than it should and we output the corresponding distribution. The second part shows that different distributions can have very similar quantile functions. We believe it shows that relying only on quantile functions makes it hard to distinguish between said distributions which is an issue for quantile based methods like QR-DQN or IQN.
> > >
> > > We are not sure of whether it is necessary to add these plots to our paper's appendix as they do not relate so much to our method or present a novelty. Which is why we first propose to add them as a supplementary material notebook. If the reviewer or editor thinks that the plots should be added to the appendix, we will do so and edit the paper further.
> > >
> > > **NFDRL-S performance**. We added the comments on NFDRL-S performance to our manuscript.
> > >
> > > We thank again the reviewer for his precious remarks.

---

> > > > ### Comment · Reviewer_XicL · 2025-06-19
> > > > **Reply to the Authors**
> > > >
> > > > Thanks for your further comments. I think it would make sense to have them in the paper (in the appendix) in the measure that you are also showing that your method does not suffer from the same issues. I find it weird that such phenomenons are known but actually never investigated or discussed to any extent anywhere to be honest... But that's it, so yes, I would prefer to have a clear reference of these claims holding in the paper personally...

---

### Review · Reviewer_QVJy · 2025-06-02

**Summary Of Contributions:**

The authors introduce a new class of distributional RL algorithms based on normalizing flow. The introduction of normalizing flow in measuring the return distribution differences is motivated to address existing algorithms' bounded support and other drawbacks. The proposed method is equipped with a variant of Cramer distance with comprehensive experiments on simple MDPs, classical control, and a few Atari games.

**Audience:**

Yes

**Claims And Evidence:**

No

**Requested Changes:**

- The efficiency of flow-based generative models lies in the high-dimensional data distribution fitting. However, oftentimes in distributional RL we only need to handle 1D data. Therefore, it is suggested to investigate the performance in the multi-variate reward function setting in RL.

- The difference between the proposed method with the Gaussian mixture model-based distributional RL should be highlighted. It seems that both methods highly rely on / are restricted by the capability of GMM.

- What if the return distribution is not continuous or smooth? As mentioned in the last paragraph in Section 3.2, the proposed method may become even worse.

- How do we analyze the approximation error of the proposed loss of Cramer distance? It is intuitively inferior to Cramer distance-based algorithm due to the existence of approximation error.

- It is unfair to compare the parameter efficiency in this way in Figure 6, as the performance of C51 also saturates quickly as the number of atoms increases. In other words, we can also expect the result in the RHS of Figure 6 for C51.

**Strengths And Weaknesses:**

### Strengths
- Introducing flow models into distributional RL is well-motivated
- The empirical demonstration is comprehensive.

### Weaknesses
- **The unboundedness benefit of using flow models is less convincing**. Firstly, one motivation of flow model is allowing unbounded support, but it suggests in Section 3.3 that it is additionally achieved by using support transformation, which seems weird and not natural. This strategy is not as advantageous as quantile-based dist RL, rendering the motivation of the new method less convincing.

- **The geometry-aware distance is not well-justified**. Although it seems novel, a new variant of Cramer distance in Section 10.1 is engineered with a sloppy and heuristic explanation in Appendix 7.3. I understand the weight calculated by the difference between two support points can induce geometry in defining a statistical distance, but I think it is less powerful than optimal transport-based distance, e.g., Wasserstein distance. Although it may involve a deep optimal transport background to point out the discrepancy or potential drawbacks of the proposed loss compared with the Wasserstein distance, it seems that it only partially utilizes the geometry information. In addition, the proof of the transition invariance property in Eq. 16 seems wrong. How does the second equation hold in general? The general proof in the literature should consider the dual property of a certain statistical distance.

- **Empirical benefits seem very marginal**. In Table 1, I cannot tell the superiority of the proposed methods as opposed to IQN. The variance is large, as pointed out in Section 6, which discourages practitioners from employing this algorithm. The authors should fundamentally modify this paper by using a more justified statistical distance when considering using a flow-based method, while conducting more experiments to demonstrate a significant performance improvement. For now, the proposed method has no advantages over IQN.

---

> ### Author Response · Authors · 2025-06-17
> **Aswer to reviewer QVJy**
>
> We thank the reviewer for these highly valuable comments that help us enhance our paper quality. We believe these crucial points deserve a thorough clarification and hereby apologise for our answer's length. Due to openreview's character limit, our answer will be split in separate consecutive comments.
>
> **The unboundedness benefit of using flow models is less convincing.** We appreciate your insightful feedback regarding the unboundedness benefit and our choice of flow architecture. We understand why our approach might initially appear less conventional than directly $\mathbb R\xrightarrow{}\mathbb R$ mapping flows like Neural Spline Flows (NSFs). However, our design is a deliberate and fundamental choice, offering distinct advantages that are crucial to our method's capabilities.
>
> Our model utilizes a flow function that maps samples from a base distribution to return values by first passing them through the CDF of a learned Mixture of Gaussians (MoG). An affine transformation with a learnable scaling parameter, $G(x,a)^{max}$, is then applied to the output of this MoG CDF. This is not simply about achieving unbounded support; it's a strategic architectural decision that provides the benefit of direct and efficient PDF Computation while leaving the door open for risk aware policies.
>
> The analytical form of the MoG CDF and its derivative (the MoG PDF) allow for direct and highly efficient computation of the target return distribution's PDF via the change-of-variables formula. Our flow's derivative is directly proportional to the PDF of the learned MoG. This explicit and analytically tractable PDF is fundamental as it uniquely enables the efficient calculation of our novel PDF-based Cramér distance surrogate. For a generic $\mathbb R\xrightarrow{}\mathbb R$ flow like an NSF, while it provides numerical access to the PDF, its internal compositional structure (a product of many Jacobian determinants) can make deriving or manipulating the precise analytical form of the overall PDF for specific loss functions (like ours, which relies on particular components of the PDF) less direct or more computationally complex to work with symbolically. Our approach offers a "white-box" component that is directly interpretable and usable within our specialized loss.
>
> While an NSF provides $p_{target}(x)$, the analytical form of this $p_{target}(x)$ is a complex, multi-layered product of Jacobian determinants. It's not a simple, interpretable parametric form like a MoG PDF. If the Cramér surrogate requires more than just the numerical value of $p_{target}(x)$ or its gradients (e.g., if it needs to analytically integrate specific parts of the PDF, like for a risk aware approach),  it would be significantly harder with the complex, composed nature of an NSF's PDF compared to the MoG-based PDF.
>
> **Addressing the Comparison to Quantile-Based Methods.** While quantile-based methods like IQN can handle unbounded supports, we argue that our flow-based approach addresses several key limitations of such methods:
> - **Continuous Density Modeling**: Unlike quantile methods that use discrete atoms, our method learns a continuous distribution with adaptive resolution, offering a more expressive and efficient representation.
> - **Convergence Guarantees**: IQN lacks convergence guarantees when using the Huber loss, whereas our Cramér-loss-based method ensures unbiased gradients and theoretical convergence via a contraction mapping.
> - **Robustness to Collapse and Quantile Crossing**: As shown in both Jullien et al. (2024a) and our own results (Figure 5), quantile-based methods often collapse toward the mean and suffer from quantile crossing; issues our approach avoids.
> - **Cramér Distance Suitability**: The Cramér distance is particularly well-suited for unbounded or non-overlapping return distributions. Quantile methods struggle to compute it efficiently due to its reliance on CDFs. Our flow model’s continuous PDF enables a novel surrogate that allows us to fully exploit this robust metric.
>
> We also refer the reviewer to our reply to Reviewer XicL regarding quantile methods’ lack of control over local spread.
>
> Finally, while we do not claim state-of-the-art performance, our aim is to provide a principled, interpretable alternative to IQN, with potential advantages in calibration, sample efficiency, and structure-aware modeling of return distributions. We hope our approach lays the ground for more elaborate models either based on normalizing flows or flow matching.

---

> > ### Author Response · Authors · 2025-06-17
> > **Aswer to reviewer QVJy (2/3)**
> >
> > **The geometry-aware distance is not well-justified**. We aknowledge that the reviewer feels that our derivation in section 7.3 is not a principled derivation. We also understand this critique as follows: ``the surrogate considers pairwise distances between support points $|y_i - y_j|$ as weights, but doesn’t explicitly solve an optimal matching problem like Wasserstein does. So while there's some geometry, it’s not “full” geometry in the OT sense."
> >
> > We thank the reviewer for raising the issue of theoretical justification. We agree that the original explanation in Appendix 7.3 was not sufficiently rigorous and appreciate the opportunity to clarify.
> >
> > In the revised version, section 7.3 now provides a more principled derivation of our surrogate geometry-aware loss. This new derivation is based on Jensen's inequality and Fubini's theorem and we show that the loss is a Riemann sum approximation of a continuous geometry-weighted loss.
> >
> > We also respectfully disagree that it only partially utilizes the geometry information. Indeed, the surrogate above is structurally related to kernel-based measures like the \emph{energy distance} and \emph{maximum mean discrepancy (MMD)} as it has a pairwise structure similar to energy distance or MMD:
> > $$E_{X \sim P, Y \sim Q}[k(X,Y)] - \frac{1}{2}E_{X,X'}[k(X,X')] - \frac{1}{2}E_{Y,Y'}[k(Y,Y')]$$
> >
> > for a geometry-inducing kernel $k(x,y)=|x-y|$.
> >
> > Therefore, our surrogate can be viewed as an inner-product kernel on the PDF difference, weighted by a geometry-aware term $|y_i-y_j|$, similar to using a linear kernel with explicit geometric structure.
> >
> > **Comparison with Wasserstein:** We emphasize that our objective is not to replicate the full structure of the Wasserstein distance, but rather to approximate its most desirable properties, namely *geometry-awareness* and *stability*,without incurring biased sample gradients.
> >
> > While the Wasserstein distance is a true optimal transport (OT) metric and exhibits contraction under the Distributional Bellman operator, its practical use in stochastic optimization is hindered by the lack of unbiased sample gradients. Additionally, computing OT metrics like Wasserstein often involves solving costly optimization problems or using approximations such as Sinkhorn divergence.
> >
> > In contrast, our surrogate loss:
> > - Is fully tractable and differentiable, relying only on the PDFs produced by the flow model.
> > - Preserves geometric sensitivity through the use of $|y_i - y_j|$ weighting.
> > - Avoids the biased sample gradient issue inherent to Wasserstein losses.
> >
> > In summary, although our loss is not a full Wasserstein metric, it retains key geometric properties while remaining differentiable with respect to model parameters. This differentiability, combined with its conceptual similarity to kernel-based methods like the energy distance and MMD, provides a practical balance between theoretical grounding, computational efficiency, and optimization stability.
> >
> > **Correctness of translation invariance**. We thank the reviewer for this very insightful remark. We totally agree that there is an issue that we address by proposing a new proof for translation invariance. The problem lies in the first line of the proof as this expression is not the direct application of our distance definition to $\pi_1^{(+b)}$ and $\pi_1^{(+b)}$ if these translated densities are still evaluated on the original support set ${y_1,\dots,y_N}$.
> >
> > We edited our proof in order to prove the property using continuous supports a more rigorous definition of translated densities. We also propose derivations for the discrete setup.
> >
> > The continuous geometry-aware surrogate distance \( d \), and its discrete counterpart \( d_{\text{disc}} \), are both invariant under translation, assuming the evaluation grid is translated consistently. This provides a principled justification for the translation invariance property of the proposed surrogate.
> >
> > **Expressivity Beyond GMMs**: While GMMs in traditional methods are used to directly model the return distribution, our approach uses them as parameterized building blocks within a flow-based transformation pipeline. The resulting distributions can go beyond mixtures of Gaussians in shape and tail behavior, especially when composed either with other CDFs or with additional transformations (such as the learnable affine scaling).
> >
> > **What if the return distribution is not continuous or smooth?** We do not say in section 3.2 that our model is limited to continuous or smooth return distribution. In figure 4(a) for instance we show that the model correctly learns a bimodal categorical distribution by assigning mass to both modes while assigning nul density everywhere else. Our model is even able to learn Diracs (figure 2(c)). Therefore the return distribution not being continuous or smooth is not an issue for our approach.

---

> > > ### Author Response · Authors · 2025-06-17
> > > **Aswer to reviewer QVJy (3/3)**
> > >
> > > **What if the return distribution is not continuous or smooth?** We do not say in section 3.2 that our model is limited to continuous or smooth return distribution. In figure 4(a) for instance we show that the model correctly learns a bimodal categorical distribution by assigning mass to both modes while assigning nul density everywhere else. Our model is even able to learn Diracs (figure 2(c)). Therefore the return distribution not being continuous or smooth is not an issue for our approach.
> > >
> > > **Multi-variate reward.** We thank the reviewer for this insightful observation. It is true that normalizing flows are often employed in high-dimensional generative modeling tasks. However, we emphasize that the strength of our approach does not lie in dimensionality per se, but in the flexibility and expressiveness that flow-based models offer, even in the univariate case.
> > >
> > > Our method leverages this expressiveness to overcome limitations of traditional quantile or atom-based DistRL methods, such as fixed support ranges, limited modality, or quantile rigidity. By using normalizing flows conditioned on state-action pairs, our model learns both the shape and the support of return distributions in a data-adaptive way, something not easily achievable with existing approaches.
> > >
> > > We agree that extending our method to multivariate reward settings is a promising direction for future work. Such settings, where returns or rewards are vector-valued (e.g., in multi-objective RL or when learning joint return distributions over time steps), would naturally benefit from the capabilities of flow-based models. We have added this discussion in the conclusion as a potential extension.
> > >
> > > **Fairness of figure 6.** The purpose of the right panel is to show how we chose the number of samples from the base distribution in our experiments. An equivalent plot for C51 would be to plot the performance of C51 depending on the number of atoms. One would then observe that the performance increases until plateauing after 51 atoms as shown in the C51 paper. We therefore insist that the point of the right panel is not to compare NFDRL to C51 but to justify our hyperparameter choice.
> > >
> > > The left panel shows that we achieve the same perfomance as C51 or better while using a fraction of its amount of parameters. We do not believe this comparison is unfair. In order to make both messages clearer we propose to separate the panels into two distinct figures.
> > >
> > > We hope that our answers have addressed your concerns and convincingly supported the validity of our claims. We are also grateful for your thoughtful feedback as your suggestions have been invaluable, and we believe they have substantially improved the clarity and quality of our work.

---

> > > > ### Comment · Reviewer_QVJy · 2025-06-20
> > > > **Reply to Authors**
> > > >
> > > > **1. The unboundedness benefit of using flow models is less convincing.**
> > > > I acknowledge that the affine transformation used in the flow seems technically sound, but I think it is still less flexible than methods that directly have an unbounded mapping, such as quantile-based methods, since we still need to define the boundary in the proposed method. It seems also to depend on the efficacy of the learned MoG to allow the analytical form.
> > > >
> > > >
> > > > **2. Addressing the Comparison to Quantile-Based Methods.**  I do not think the Huber loss, the robust version of L2 loss, cannot lead to the convergence guarantee. I am emphasizing the general quantile-based algorithm, such as QR-DQN, where the convergence of dynamic programming is well-established. Also, the non-crossing issue does not affect the performance of quantile-based methods too much based on experience. Addressing this issue can even make the performance better, while it is disappointing to see that the performance of the proposed method is not even as competitive as IQN.
> > > >
> > > >
> > > > **3. The geometry-aware distance is not well justified.** The kernel-based distance, like MMD, cannot fully capture the data geometry compared with optimal transport distance. Therefore, the weight function captures some geometrical information, but not as much as optimal transport distance, such as Wasserstein distance. An intuitive explanation is that optimal transport distance considers the joint pdf in moving mass by definition, while here you only consider a pairwise weight function.
> > > >
> > > >
> > > > **4.Empirical performance.** The authors emphasize many times that the advantage of the proposed method lies in the flexibility and expressiveness, but I am disappointed to find that the empirical results cannot support this claim. Thus, it is much less convincing to persist on this claim.

---

> > > > > ### Author Response · Authors · 2025-06-21
> > > > > **Reply to reviewer QVJy**
> > > > >
> > > > > We thank the reviewer for their thoughtful follow-up. We understand that the absence of previously raised concerns in the latest response suggests that those points are now considered adequately addressed. If there are any remaining issues that still require clarification, we would be happy to revisit them.
> > > > >
> > > > > **need to define boundary**: We respectfully disagree as the boundaries are not defined by the user but learnt and can be arbitrarily large.
> > > > >
> > > > > **MoG efficacy**: We agree that the expressiveness and quality of the learned distribution directly impact the performance of the model. However, this dependency is intrinsic to any parametric generative model, much like how the effectiveness of neural networks depends on the quality of the learned weights. We also show in figure 5 how our method is more precise and flexible than IQN, as the latter fails to precisely learn a bimodal distribution while our method does.
> > > > >
> > > > > Moreover, We maintain that our method presents richer capacity than quantile based methods and point the reviewer to our answer to reviewer XiCL regarding quantile methods’ lack of control over local spread.  We believe that dependency to the MoG efficacy does not constitute a limitation, but rather reflects the natural behavior of any learned probabilistic model.
> > > > >
> > > > > **The QR-DQN convergence of dynamic programming is well-established.** We kindly disagree as the authors of QR-DQN and IQN state it themselves. In the conclusion of the IQN paper the authors ask: ``sample-based convergence results have been recently shown for a class of categorical distributional RL algorithms (Rowland et al., 2018). Could existing sample-based RL convergence results be extended to the QR-based algorithms?".
> > > > >
> > > > > This issue is even more serious for IQN as in the same section they ask: ``Can the contraction mapping results for a fixed grid of quantiles given by Dabney et al. (2018) be extended to the more general class of approximate quantile functions studied in this work?"
> > > > >
> > > > > To summarize,
> > > > > - there is no sample based convergence proof for quantile based method as opposed to PDF based ones
> > > > > - There is even no proof for the contraction mapping using IQN
> > > > >
> > > > > Although these model show great empirical performance, as long as these proofs are missing, we consider that there is no performance or even convergence guarantees for these models compared to ours.
> > > > >
> > > > > **Non-crossing issue.** On this issue we cite Zhou et al. in our paper. Their figure 1 shows how crossing quantiles affect the learnt quantile function and action selection. Their model learns smoother functions and increase accuracy. However this is not the case for IQN as they state it themselves: ``the implementation approach in this paper can not be directly applied to some distribution based methods, such as IQN, since the quantile fractions $\tau$ ’s are not fixed and re-sampled each time".
> > > > >
> > > > > Finally, we do not agree that it does not have too much impact on performance. We point to table 1 of Zhou et al. that shows that the mean performance is nearly doubled compared to standard QR-DQN. As this issue is still pending for IQN, and was shown to be detrimental in our figure 5, we believe that the impact on IQN performance should not be negligible.
> > > > >
> > > > > **No as competitive as IQN.** We share the reviewer's disappointment but we insist again that we do not aim for state of the art performance but to offer a principled method that:
> > > > > - avoids the issues of Wasserstein distance while keeping its beneficial properties
> > > > > - offers all the convergence and contraction proofs that lack for quantile based methods as said above.
> > > > >
> > > > > We agree that IQN outperforms our model on some cases but there are still many cases where our own model is close. However, given the our model's simplicity, we believe it is still possible to achieve better performance using different flow functions, which will be explored in future work.
> > > > >
> > > > > Finally we would like to kindly remind the reviewer that TMLR seeks to advantage soundness and correctness over empirical performance which we believe our paper does.
> > > > >
> > > > > **kernel-based distance.** We agree but we only emphasize that our surrogate bears the same benefits as MMD. Moreover, one can compute Cramér distance (although with less computational efficiency), therefore this argument can only hold against our surrogate but not the overall method as it offers the opportunity to exactly compute the Cramér distance.
> > > > >
> > > > > **flexibility and expressiveness.** We maintain our claim as we believe all the cited issues of existing methods cited above coupled with the qualitative advantages cited for our method are a good explanation of the behaviour illustrated in Figure 5.

---

> > > > > > ### Comment · Reviewer_Wrmx · 2025-06-21
> > > > > > **Just butting in regarding a point**
> > > > > >
> > > > > > Hi,
> > > > > >
> > > > > > I just wanted to in to make a note regarding the authors' statement "there is no sample based convergence proof for quantile based method as opposed to PDF based ones". This was proven in the 2023 JMLR paper "An Analysis of Quantile Temporal-Difference Learning". With that in mind, I hope the authors can adjust their statements.

---

> > > > > > > ### Author Response · Authors · 2025-06-26
> > > > > > > **Answer to reviewer QVJy on quantile methods convergence**
> > > > > > >
> > > > > > > We would like to thank the reviewer for suggesting this highly relevant paper that we were not aware of unfortunately. It does prove very important results for quantile based methods and definitely has to be cited in our own article. However, after its careful study, we are able to maintain our stance.
> > > > > > >
> > > > > > > Indeed, its main theorem (Theorem 8) proves that: The iterates of the QTD algorithm converge almost surely to the set of fixed points of a projected distributional Bellman operator (depending on quantile interpolation), under standard stochastic approximation assumptions.
> > > > > > >
> > > > > > > While this convergence result is strong and addresses prior theoretical gaps, note the following:
> > > > > > > - It is specific to tabular QTD (i.e., no function approximation or deep networks).
> > > > > > > - It shows convergence to a fixed point of a projected operator, not necessarily to the true return distribution.
> > > > > > > - It does not directly apply to QR-DQN or IQN, which sample from a continuous distribution of quantiles and use function approximation (e.g., neural networks). They remain without a full convergence proof, especially under deep function approximation.
> > > > > > >
> > > > > > > This was also noted by the authors themselves in the paper's conclusion: "Another important direction is to analyse more complex variants of the QTD algorithm, incorporating more aspects of the large-scale systems in which it has found application. Examples include incorporating function approximation...".
> > > > > > >
> > > > > > > Therefore we maintain our claim but we agree that it requires the following precision: there is no sample based convergence proof for quantile based methods *that make use of function approximation like QR-DQN or IQN*.
> > > > > > >
> > > > > > > Anyway, we appreciate the reviewer's suggestion and thank  him for letting us know of this important work. We added all these precisions in a new revision.  (Please do not hesitate to delete your browser's cookies in case you cannot view the changes made in the introduction)

---

### Review · Reviewer_Wrmx · 2025-06-11

**Summary Of Contributions:**

The paper introduces a new method named NFDRL, a distributional RL architecture which aims to represents return distributions with normalizing flows rather than existing parametric families. They derive a way to train these models, and make use of a surrogate Cramér distance through the use of Taylor approximations, in order to compute it on PDFs. They perform empirical evaluation on a collection of toy MDPs and the Atari-5 benchmark, and provide comparative visualizations of the learnt distributions, as well as the returns achieved by agents using this method.

**Audience:**

Yes

**Broader Impact Concerns:**

I don't think there are any ethical concerns.

**Claims And Evidence:**

No

**Requested Changes:**

- Based on the above, I think there are some major modifications needed in Section 7.3.2 in order to have a contraction proof.
- Similarly, I think it is necessary to formalize and clean up the log-likelihood derivation, and have some way of formally justifying each step to the reader.
- I disagree with the statement "It also offers richer modeling capacity to capture multi-modality, skewness, and tail behavior than quantile based approaches." in the abstract, and I believe this to be unqualified based on the results in the paper. Quantile-based approaches have no issues representing these properties (the quantile projection which preserves all of these properties).
- I believe the authors should either generalize their proof of unbiased sample gradients to general distributions, or make it clear that their claim only applies to Bernoulli distributions.


Minor nits:
- The use of $\pi_1,\dots$ for PDFs over $\mathbb{R}$ is a bit jarring when $\pi$ is used to represent a policy in the rest of the paper (and indeed this is the general usage of this notation in RL).
 - Figure 1 caption: "bootsrap" -> bootstrap
- I believe it should be Cramér instead of Cramèr.

**Strengths And Weaknesses:**

**Strengths**
- Representing return distributions via normalizing flows is an interesting and new idea.
- The Taylor approximation of the Cramér distance is a nice idea and might be useful in distributional RL more generally.
- I appreciate the authors being upfront regarding the limitations of their approach in Section 6, I think this academic honesty is valuable and will lead to simpler paths for future work.

**Weaknesses**
- I think there are some issues with the derivations in Section 7.3.2. Firstly, I do not see why $\pi_1^{(a)}$ is a proper PMF? Following the exact definition, if $\pi_1$ puts equal probability on $\\{ 1, 2 \\}$, and the scaling is $a=10$, then $\pi_1^{(a)}$ puts probability $1/20$ on each of $ \\{ 10, 20 \\}$, which is clearly not a PMF. If that is resolved, I believe that the inequality under the line "Therefore, the metric d satisfies the scaling property:" is going the wrong direction based on the preceding argument, which of course is the opposite direction needed to guarantee a contraction.
- The authors claim that their surrogate distance $d$ has unbiased sample gradients, but their proof in Section 7.3.3 is only in the setting of  Bernoulli distributions.
- The mapping from CDF to output described in Section 3.3 does not seem correct in general and I am a bit confused by its motivation. It seems that $f(y)$ has the goal to be an approximate inverse of the CDF $F$, but the form they use is an exact inverse in the case of uniform distributions, but it is not exact in general.
- Similar to the above, I find it hard to follow and understand the motivation for including this term in the log-likelihood, and similarly the terms in the log-likelihood of (7) and (8). I think if the authors can generally be more precise and clear when deriving the log-likelihood that would help sort out potential issues (it's not clear to me that the log likelihood as is is well-defined, since from my understanding the CDF is being treated as a likelihood function).

---

> ### Author Response · Authors · 2025-06-17
> **Answer to reviewer Wrmx**
>
> **Issue section 7.3.2**. We thank the reviewer for spotting this mistake that is more a writing issue than a methodological one. Indeed, the formulation "Let $\pi_1$ and $\pi_2$ be two probability density functions over the support $\{y_1,\dots, y_N\} \subset \mathbb R$" can imply that the distribution is defined over a discrete support and is therefore a PMF.
>
> However as stated above, we define $\pi_1$ and $\pi_2$ as PDFs rather that PMFs and we should have adopted the same formulation as the one used in the triangle inequality proof: "Let $\pi_1$ and $\pi_2$ be two probability density functions **evaluated** over the support $\{y_1,\dots, y_N\} \subset \mathbb R$"
>
> This way $\pi_1$ and $\pi_2$ are two PDFs defined on a continuous support but evaluated on a finite set of points. Indeed, as we use a finite set of samples to evaluate each return distribution this formulation is more in line with what we do in practice while preserving the fact that $\pi_1$ and $\pi_2$ are PDF and the correctness of the subsequent proofs. We edited the article accordingly.
>
> **Missing proof for section 7.3.3** Thank you for noticing this mistake, it seems that the proof for general distributions disappeared from our manuscript. We apologise for this issue. We edited the manuscript with the correct general proof. The Bernoulli distribution is only used for illustration purpose.
>
> **Clarifications on section 3.3**. The flow function takes the form of a CDF, therefore when $z$ is sampled from the base distribution and input in the flow function $F$, the output $F(z)=y$ is in $[0,1]$.
>
> However, $y$ is supposed to be a sample from the return distribution and hence should not be bounded in. By inputing $y$ into a function $f$ defined as $f(y) = 2\cdot y \cdot G^{max} - G^{max}$, $f(y)$ now takes values in $[-G^{max},G^{max}]$.
>
> The neural network $h_\theta$ outputs the right value $G^{max}$ depending on the input state, making $f(y;s,a)$ virtually unbounded.
>
> Normalizing Flows are based on the change of variable formula to get the exact likelihood of any sample from the target distribution. Before the scaling operation the likelihood of $y$ under the return distribtion is:
> \begin{equation*}
> \log \eta^\pi(x,a)(y) = \log p_{\mathcal{U}}(z) - \log \left| \frac{dF_\theta(z)}{dz} \right| .
> \end{equation*}
>
> The function $f$ rescales the predicted return distribution, therefore its log-derivative has to be substracted from the likelihood above. The derivative of $f$ being $2G^{max}$, we get:
> \begin{equation*}
> \log \eta^\pi(x,a)(y) = \log p_{\mathcal{U}}(z) - \log \left| \frac{dF_\theta(z)}{dz} \right| - \log |2\cdot G^{max}|.
> \end{equation*}
>
> Based on these elements, we can conclude that $f(y)$ does not have the goal of being the approximate inverse of the CDF but a tool to expand the support of the CDF beyond $[0,1]$. Although the result is not a CDF anymore, it keeps its convenient properties (diffeomorphism).
>
> **Clarifications on the added terms in the likelihoods**. We thank the reviewer for his advice and we edited the manuscript to link more clearly equations 7,8,9,10 to equation 6.
>
> We recalled in section 2 that starting from a base sample $z_0 \sim p(z_0)$, a flow applies a sequence of transformations $f_k \circ \dots \circ f_1$ to obtain $z_K$, whose density is computed using the change of variables formula:
> \begin{equation*}
> \log p_\theta(z_K) = \log p(z_0) - \sum_{k=1}^K \log \left| \det \left( \frac{\partial f_k}{\partial z_{k-1}} \right) \right|.
> \end{equation*}
>
> Said otherwise, if flows are composed one after the other, it suffices to substract the log jacobian of each added flow to the previous likelihood to get the new exact likelihood.
>
> This is exactly what was done in section 3.3 where the function $f$ is considered as a flow function that comes after the first CDF flow $F$. This is the reason why we substracted the log of f's derivative.
>
> The same reasoning is applied for equation 7, where an additional flow has been used after $f$, defined as $g(y) = r + \gamma y$ and used to build the RL target distribution. its derivative is $\gamma$, hence the fact that we substracted $\log(\gamma)$ in equation 8.
>
> We also recognize that the notation can be misleading as we used $y$ to denote the output of both $F$ and $f$ in section 3.3 and then used again $y$ to denote the output of $g$ in section 3.4. The rational was that the final return sample should be $y$ and this sample was being updated after each flow while keeping the notation $y$. The notation was clarified in the updated version to introduce $\tilde{y}$, the output from the target flow earlier.
>
> **Richer capacity than quantile based methods**.  We refer the reviewer to our reply to Reviewer XicL regarding quantile methods’ lack of control over local spread. Moreovoer, as shown in both Jullien et al. (2024a) and our own results (Figure 5), quantile-based methods often collapse toward the mean and suffer from quantile crossing; issues our approach avoids.

---

> > ### Comment · Reviewer_Wrmx · 2025-06-22
> > **Reply to authors**
> >
> > **Note**: I see the authors have stated that they made revisions and I see a revised PDF was uploaded, but none of the stated revisions regarding my points have been made. I'd ask the authors to double-check that it was successfully uploaded so that I can properly see their changes. Also, when the revisions are uploaded (if they aren't already), I would strongly suggest the authors to make all revised text in a different colour to help the reviewers.
> >
> > **7.3.2**: I appreciate the authors agreeing to make the change so that the PDFs have continuous support, I can believe that the definition is valid in that case. However, the more important point which is not addressed in the response is the direction of the inequality in the "Scaling" section. I would request that the authors double-check this, and if they still believe it to be true to include a proper derivation of the inequality to convince readers of the fact beyond doubt.
> >
> > **7.3.3:** As stated above the proof is still only in the Bernoulli case, so I can't comment on the authors' response as the current PDF contradicts their statement "We edited the manuscript with the correct general proof".
> >
> > **Section 3.3:** I am unfortunately not convinced by the authors current arguments, and as before I'd ask that the derivation of the normalizing flow be more explicitly made clear in the paper, in order to validate its correctness. For example, I would assume that if an input sample is transformed into a CDF, the probability which is implictly being learnt is $P(F_{Z} = a)$. Similarly I don't believe that the authors' argument for the map $f(y)$ is a reasonable map to use -- indeed the "true" map which should be used is the inverse CDF (since this is exactly what maps from CDF values to the sample which attains that value). If they want to argue that their proposed $f(y)$ is a useful surrogate to this function (which I don't currently believe it is and is why I am inviting the authors to defend this point), I would expect a stronger argument than simply that $f$ maps $[0,1]$ into $[-G_{max}, G_{max}]$.
> >
> >
> > **Richer capacity than quantile based methods**: I stand with my original point, and I believe the authors have to be more clear with their statements. For example saying "quantile-based methods often collapse toward the mean and suffer from quantile crossing" is much too broad -- of course some quantile-based implementations suffer from quantile crossing (e.g. QR-DQN, IQN), yet some don't (e.g. NC-QR-DQN (Zhou et al., 2020), SPL-DQN (Luo et al., 2023)). With this context the statement quantile-based methods suffer from quantile crossing is therefore unfounded. The exact same can be said about "often collapse toward the mean" -- the true fixed-point distribution of quantile distributional RL methods has no issues with collapsing to the mean, and the issue for mean-collapse brought forward by Jullien et al. (2024a) addresses the particular case when the Huber loss is used instead of the standard quantile regression loss. All in all, I believe that there are advantages of this work over quantile methods (and categorical too), but the current argument isn't accurate and can mislead future work which might try to build off this.

---

> > > ### Comment · Action_Editor_6cYQ · 2025-06-24
> > >
> > > Dear authors,
> > >
> > > I'd like to follow up on the comment by the reviewer that the PDF is not revised as promised. Will you upload a new version?
> > >
> > > Thank you,
> > > Action Editor

---

> > > ### Author Response · Authors · 2025-06-25
> > > **Answer to Wrmx**
> > >
> > > We are quite surprised that our revisions are not accessible as other reviewers do not seem to have this issue. We added a new revised manuscript where all the revisions have been highlighted in red. If this issue persists we kindly ask the reviewer to delete his cookies before trying to download the document again.
> > >
> > > **Scaling issue.** We thank the reviewer for noticing this issue and also apologise for misunderstanding the his first comment. It is totally right that the scaling proof was wrong and it is true that it implies that the contraction proof is wrong too. We apologise for this unfortunate mistake.
> > >
> > > We propose to solve that by proposing a new surrogate loss that elevates the $|y_i-y_j|$ to the power 2. The new surrogate loss is therefore:
> > > $$\mathcal{L}^2(p, q) = \frac{1}{N^2} \sum_{i=1}^N \sum_{j=1}^N (p(y_i) - q(y_i))^2 \cdot |y_i - y_j|^2$$
> > >
> > > We argue that this loss while not approximating the Cramér distance keeps all its benefits while remaining closely related to energy distances and can be viewed as an inner-product kernel on the PDF difference.
> > >
> > > We have done again all the proofs and proved that this new loss is a true distance, it is scale and translation invariant. We also show that it is a $\gamma$-contraction for the Bellman operator and that the sample gradients are unbiased. All the new content is shown in red.
> > >
> > > We also started new experiments to evaluate this new loss. However we are writing to editor to ask for the rebuttal to be extended by a couple of days in order to be able to finish all evaluations. From our first experiments we observe that the performance seems similar to the previous surrogate, however we also observe less variance during the learning process. These observations have to be confirmed while experiments on all ATARI-5 games. Relevant figures and tables will be updated upon experiments completion.
> > >
> > > **Section 3.3**. There seems to be a misunderstanding about the principle of normalizing flows and how they are used in our paper. The CDF function is not used to learn a probability. It is only used as a function that maps a input sample to [0,1]. The corresponding likelihood is learnt through the change of variable formula (equation 6).
> > >
> > > To illustrate, consider that he CDF can be replaced by any other invertible function $f$. This function maps an input $z$ to $x$ in a certain domain $[a,b]$. Similarly to activation functions for neural networks, an additional function $g$ can be used on top of the function $f$, mapping $x$ to $y$, that changes the support from $[a,b]$ to $[c,a]$. We have $y=g(x)=g(f(z)$. This function $g$ can even be a ReLu function for example.
> > >
> > > In this case the function $f$ is not a CDF but the change of variable would still be valid. The likelihood of $y$ can be computed from equation 6:
> > > $$\log p_\theta(y)=p(z)-\log \frac{\partial f}{\partial z} - \log \frac{\partial g}{\partial x}$$
> > >
> > > In our approach, we chose to model the function $f$ as a CDF parameterized by a learnt mixture of gaussians and the function $g$ as a learnt affine function. The fact that our flow function is a CDF does not mean in any way that we assume that we learn the CDF of the return distribution. We learn the PDF of the return distribution using a flow function that takes the form of a CDF that is not in any case the CDF of the learnt distribution.
> > >
> > > Finally, using the inverse CDF after the CDF would put us back at the base distribution sample which obviously is not what we want.
> > >
> > > **Huber loss.** We respectfully disagree that using the Huber loss is a particular case of using the quantile regression loss. The Huber loss is the loss used by default in both QR-DQN and IQN as stated in their respective articles.

---

> > > > ### Author Response · Authors · 2025-06-27
> > > > **New results**
> > > >
> > > > We follow up on our previous answer to inform the reviewer that we have edited our paper with the new results as well as the supplementary. The new results are displayed in table 1 and show a slightly better but very close performance to our first model. We did not update figures 2,3 and a4 as we obtained very similar qualitative results with the new loss.
> > > >
> > > > It is also to be noted that the high variance mentionned in the limitations section seems significantly mitigated and we believe this is due to the new correct contraction result. However, the observed variance remains higher than what can be observed using C51 for instance. Moreover, due to the time constraint we did not fine tune our model and used the same hyperparameters as we used before and cannot comment the model's sensitivity to said parameters. Therefore we prefer to be transparent on the possible high variance that can be observed and keep the limitations section as it is.
> > > >
> > > > We thank the reviewer and editor for their help.

---

### Decision · Action_Editor_6cYQ · 2025-08-13

**Recommendation:** Reject

**Additional Comments:**

The paper proposes using normalizing flows to represent return distributions in distributional reinforcement learning, with a motivation that it can handle unbounded returns, which are relevant in certain applications.

Two reviewers (both experts in distributional RL) recommend *leaning reject*, and one recommends *leaning accept* (expert in RL, but not distributional RL). The reviewers generally agree the idea of using normalizing flow in the context of distribution RL is interesting and has promise, but they believe the work needs substantial improvement before publication.

Several concerns have been raised. I mention some of them:

- **Theoretical correctness and clarity:** Some of the concerns have been about the theoretical justification and correctness of the method. The authors revised their paper in order to fix some of them. It is noteworthy that one of their changes had substantial consequences and required changing parts of the formulations of the method, re-proving some theoretical results, and re-running the experiments.

- **Empirical performance:** The method did not demonstrate an empirical advantage over strong baselines, and in some cases underperformed. The claimed benefits such as handling unbounded returns and geometry awareness were not shown in experiments.

- **Representation of prior work:** There have also been concerns about misrepresenting of the prior work.

In my own reading of the revised paper, I found that some steps are not as clearly explained as they could be, and certain components of the method feel like heuristic or ad hoc design choices rather than being grounded in a fully principled justification.

Given that some theoretical and empirical concerns remain, the method underwent substantial changes during rebuttal without all reviewers re-verifying all the details, and two of the three reviewers recommend rejection, unfortunately I cannot recommend acceptance of this paper at this stage. A further round of careful revision and review is needed.

**Audience:**

Yes

**Audience Explanation:**

The topic of this paper is very relevant to the RL researchers, especially those interested in the distributional RL.

**Claims And Evidence:**

No

**Claims Explanation:**

There are issues with the claims. Please read the full report.

**Resubmission Of Major Revision:**

The authors may consider submitting a major revision at a later time.